# SpecPL: Disentangling Spectral Granularity for Prompt Learning

**Jingtao Zhou** [* 1] **Xirui Kang** [* 1] **Feiyang Huang** [* 1] **Lai-Man Po**[† 1]

## Abstract

Existing prompt learning for VLMs exhibits a modality asymmetry, predominantly optimizing text tokens while still relying on frozen visual encoder as holistic extractor and neglecting the spectral granularity essential for fine-grained discrimination. To bridge this, we introduce Disentangling Spectral Granularity for Prompt Learning (SpecPL), which approaches prompt learning from a novel spectral perspective via Counterfactual Granule Supervision. Specifically, we leverage a frozen VAE to decompose visual signals into semantic low-frequency bands and granular high-frequency details. A frozen Visual Semantic Bank anchors text representations to universal low-frequency invariants, mitigating overfitting. Crucially, fine-grained discrimination is driven by counterfactual granule training: by permuting high-frequency signals, we compel the model to explicitly distinguish visual granularity from semantic invariance. Uniquely, SpecPL serves as a universal plug-and-play booster, revitalizing text-oriented baselines like CoOp and MaPLe via visual-side guidance. Experiments on 11 benchmarks demonstrate competitive state-of-the-art performance, achieving a new performance ceiling of 81.51% harmonic-mean accuracy. These results validate that spectral disentanglement with counterfactual supervision effectively bridges the gap in the stability-generalization trade-off. Code is released at https://github.com/Mlrac1e/SpecPL-Prompt-Learning.

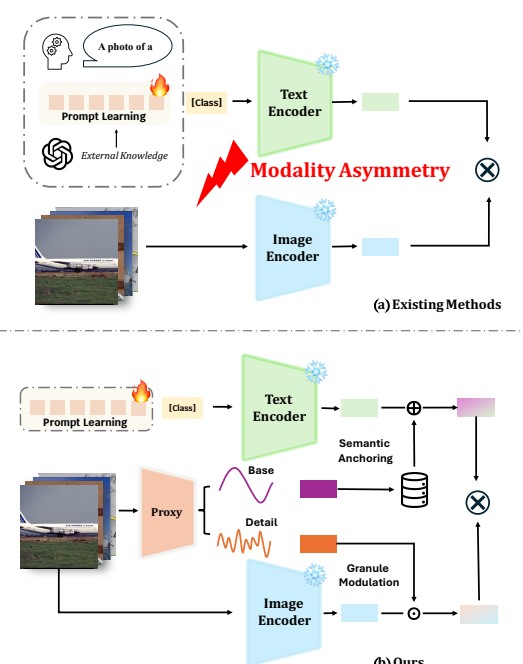

*Figure 1.* **Breaking the Modality Asymmetry.** **(a) Existing Methods** suffer from severe *modality asymmetry*: optimization is heavily concentrated on the textual side (often augmented by noisy external knowledge like LLMs), while visual representations remain **holistic** and static (indicated by the red bolt). **(b) Ours (SpecPL)** bridges this gap via *spectral disentanglement*. We introduce a **Spatial-Spectral Proxy** to decompose visual signals into **Base** (semantic) and **Detail** (granular) components. Crucially, SpecPL implements a dual-path interaction: 1) **Semantic Anchoring** (⊕): Base components retrieve invariant prototypes from the **Visual Semantic Bank** to anchor text representations; 2) **Granule Modulation** (⊗): Detail components explicitly modulate visual features to inject discriminative granularity. This ensures a balanced alignment without relying on external priors.

## 1. Introduction

Pre-trained vision–language models (VLMs) such as CLIP (Radford et al., 2021) have become a default backbone

for low-shot transfer by aligning images and texts in a shared embedding space. A dominant adaptation paradigm is prompt learning, which freezes the backbone and only updates a small set of prompt parameters. Despite its practicality, existing prompt learning methods (Zhou et al., 2022b;a; Khattak et al., 2023b; Ren et al., 2023; Tian et al., 2024; Yao et al., 2024; Zhu et al., 2024; Hua et al., 2025; Pan et al., 2025; Zheng et al., 2025a; Li et al., 2025) exhibit a persistent *modality asymmetry*: optimization is concen-

---

[*]Equal contribution [†] Corresponding author. [1]Department of Electrical Engineering, City University of Hong Kong, Hong Kong SAR. Correspondence to: Lai-Man Po <eelmpo@cityu.edu.hk>.

*Proceedings of the 43rd International Conference on Machine Learning*, Seoul, South Korea. PMLR 306, 2026. Copyright 2026 by the author(s).

trated on text tokens, while visual representations are often treated as fixed, holistic features. This asymmetry makes it difficult to elicit the hierarchical granularity needed for fine-grained discrimination, and it frequently manifests as a stability–generalization tension: prompts overfit the scarce training distribution while compromising the transferable invariances encoded by the frozen VLM.

A popular response is to strengthen the textual side by importing external knowledge, most notably using large language models (LLMs) to generate class attributes or detailed descriptions (Tian et al., 2024; Li et al., 2025; Zheng et al., 2024; Khattak et al., 2025). These LLM outputs are then treated as supervision signals, initialization priors, or auxiliary constraints. While effective in many cases, this direction introduces new costs and failure modes: generations are noisy (Roth et al., 2023) and dataset-dependent; reliance on an external model increases computation; and the resulting training signal remains text-centric, further entangling adaptation with modality asymmetry. More fundamentally, when fine-grained supervision is injected primarily through text, the visual side still lacks a principled, controllable mechanism to provide discriminative granularity.

In parallel, visual prompting attempts (Jia et al., 2022) to mitigate this issue by injecting learnable tokens into the vision encoder. However, these approaches are typically formulated at the spatial/patch-token level and lack an explicit mechanism to separate semantic invariants that define categories from fine-grained cues that vary across instances, including texture patterns, illumination conditions, and pose changes. Without this separation, optimizing the visual side can over-amplify nuisance variations, making adaptation unstable and exacerbating the stability–generalization trade-off.

**Revisiting Spectral Analysis in Deep Manifolds.** We posit that the missing inductive structure is spectral, but it must be reinterpreted in deep representations. Classical signal processing studies frequency components in pixel space; however, directly transferring this notion to semantic learning is non-trivial because pixel-level frequencies are not aligned with semantic factors. Motivated by the manifold hypothesis, we view spectral granularity in a spatially aligned latent manifold (Rombach et al., 2022) induced by a frozen generative encoder such as a VAE. In this representation space, low spatial frequencies capture slowly varying, category-level structures that remain stable across instances, including coarse shape, layout, and topology, whereas high spatial frequencies encode rapidly varying details such as textures and other instance-specific factors. Building on this view, we argue that efficient VLM adaptation benefits from a dual-band mechanism: low-frequency invariants provide a stable anchor for global semantic alignment, while high-frequency details supply discriminative cues that sharpen

fine-grained distinctions within a category. Accordingly, we instantiate *spectral decomposition* not as an explicit Fourier transform, but as a spatial-frequency factorization directly in the latent space. This reinterpretation enables an efficient disentanglement of semantic invariants from granular details, yielding adaptation that is simultaneously stable at the global level and discriminative at the local level.

To this end, we propose **Disentangling Spectral Granularity for Prompt Learning (SpecPL)**, which reframes prompt learning as *spatial-spectral factorization with counterfactual supervision*. SpecPL introduces a frozen VAE teacher to decompose each image into two complementary components via a Spatial-Spectral Proxy: a "Base" component (low-frequency proxy) capturing stable semantic invariants and a "Detail" component (high-frequency proxy) encoding granular textures. We use the Base component to *anchor* adaptation: a frozen *Visual Semantic Bank* stabilizes text representations by aligning them to universal, low-frequency invariants, mitigating overfitting.

Crucially, we use the high-frequency component to drive *fine-grained discrimination* via **Discriminative Granule Supervision**. Recognizing that VLMs often exhibit a "shape bias" that overlooks subtle textures, we introduce an identity-swapping mechanism: by permuting high-frequency signals across samples and enforcing prediction of the **granule source**, we compel the model to acknowledge that **granular details are sufficient to alter class identity**. This forces the visual encoder to actively encode high-frequency evidence rather than treating it as negligible noise.

SpecPL is designed as a universal, plug-and-play booster. It can be integrated into text-oriented baselines (e.g., CoOp, CoCoOp, MaPLe, and MMRL) with minimal intrusion: the VAE teacher remains frozen, and the additional objectives serve as training-time guidance. Importantly, inference does not depend on the VAE, enabling efficient deployment. Extensive experiments on 11 benchmarks demonstrate that SpecPL consistently improves both stability and generalization, achieving state-of-the-art performance and strengthening the harmonic-mean accuracy in base-to-novel evaluation.

In summary, our contributions are threefold:

- We identify the lack of spectral granularity factorization as a key bottleneck behind modality asymmetry in prompt learning;
- We introduce a **Spatial-Spectral Proxy** mechanism that effectively disentangles semantic invariance from discriminative details within a frozen VAE manifold;
- We provide a practical plug-and-play framework with counterfactual supervision that revitalizes existing baselines without relying on LLM-generated priors.

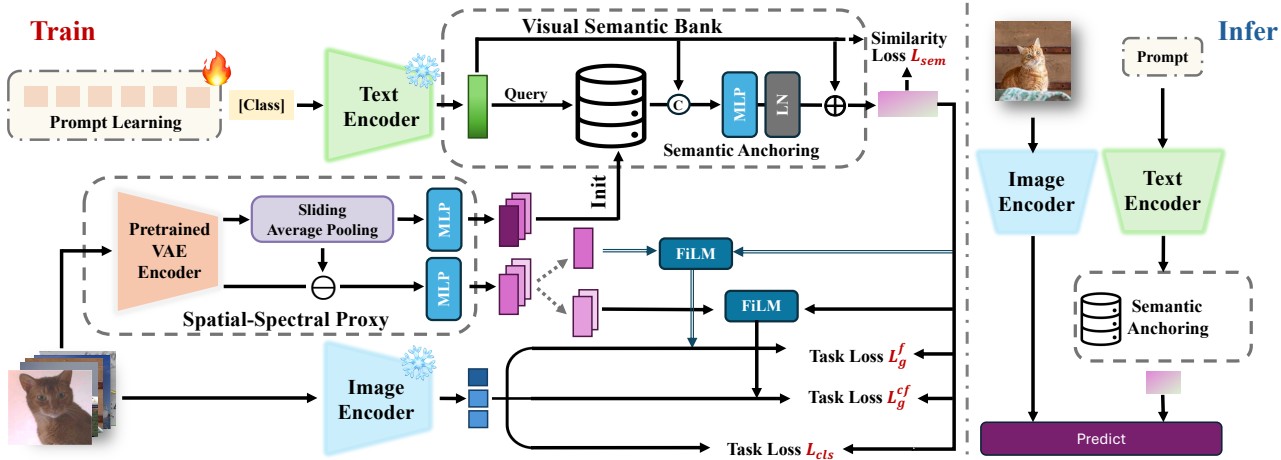

*Figure 2.* **SpecPL framework (train vs. inference).** Given an input image, a *frozen* pretrained VAE encoder provides spatial latents that are factorized by a lightweight *Spatial–Spectral Proxy* (sliding average pooling) into a low-frequency base band (semantic invariants) and a high-frequency residual detail band (instance granules). Two lightweight projection heads map the two bands into the CLIP embedding space, producing $\mathbf{t}^{\text{low}}$ and $\mathbf{t}^{\text{high}}$; the CLIP backbone remains frozen while prompts and lightweight modules are optimized. The low-band prototypes initialize a *Visual Semantic Bank* and enable *compositional text refinement* via soft retrieval followed by an MLP+LN residual aggregator, yielding refined class text features used for the main classification loss $\mathcal{L}_{\text{cls}}$. A *semantic anchoring* loss $\mathcal{L}_{\text{sem}}$ aligns the expected text direction with $\mathbf{t}^{\text{low}}$ (stop-gradient through pseudo-labels). **Training only**, SpecPL adds a FiLM-style modulation branch conditioned on fused shared semantics and $\mathbf{t}^{\text{high}}$ to impose factual and counterfactual granule supervision ($\mathcal{L}_g^f$, $\mathcal{L}_g^{cf}$). **Inference** discards the VAE teacher and FiLM branch, relying solely on bank-based refinement and standard CLIP similarity for prediction.

## 2. Related Work

### 2.1. Efficient Adaptation of Vision-Language Models

Large-scale pre-trained VLMs like CLIP (Radford et al., 2021) and ALIGN (Jia et al., 2021) have demonstrated zero-shot capabilities. However, adapting them to downstream tasks with minimal overhead remains a core challenge.

**Textual Prompt Learning.** Inspired by NLP, CoOp (Zhou et al., 2022b) introduced learnable soft prompts to replace hand-crafted templates. Subsequent works like Co-CoOp (Zhou et al., 2022a) and MaPLe (Khattak et al., 2023a) improved generalization by conditioning prompts on image instances or coupling vision-language branches. Recently, methods such as ProDA (Lu et al., 2022) and CuPL (Pratt et al., 2023) leverage external LLMs to generate auxiliary descriptors. While effective, these text-centric approaches typically treat the visual input as a **holistic embedding** and do not explicitly inspect or modulate fine-grained visual structure, which can limit discrimination in tasks requiring granular distinction.

**Visual Adaptation and Prompting.** Parallel efforts focus on the visual side. CLIP-Adapter (Gao et al., 2024) and Tip-Adapter (Zhang et al., 2021) employ residual layers or cache-based retrieval atop visual features. Visual Prompt Tuning (Jia et al., 2022) injects learnable tokens into the vision transformer. Although these methods operate on visual representations, they are typically formulated at the **spatial or token level** and do not explicitly disentangle

*semantic invariance* from *granular details*, which can lead to optimization instability in fine-grained regimes.

### 2.2. Spectral and Granularity Dynamics in Deep Representations

Visual understanding inherently involves multi-scale analysis. Classical approaches utilize Fourier or wavelet transforms where low frequencies capture global shape and high frequencies encode local texture (Geirhos et al., 2018; Wang et al., 2020). However, applying pixel-level frequency analysis to semantic tasks is hampered by the semantic gap.

Recent studies have shifted focus to the spectral properties of *deep feature manifolds* (Raghu et al., 2021). Drawing on Slow Feature Analysis (SFA) (Wiskott & Sejnowski, 2002), a common view is that deep encoders learn spatially-aligned latent spaces with spectral bias: slowly-varying components correspond to structural semantics, while rapidly-varying fluctuations capture instance-specific details or aliasing artifacts (Zhang, 2019).

**SpecPL Position.** Our work bridges efficient VLM adaptation and latent spectral analysis. Unlike prompt learning methods that treat vision holistically, and unlike frequency methods that operate on pixels, SpecPL implements a **Spatial-Spectral Proxy** directly within a frozen VAE manifold. By formulating granularity disentanglement as a spatial factorization and enforcing **sensitivity** via **counterfactual supervision**, we introduce a controllable fine-grained visual structure for prompt learning.

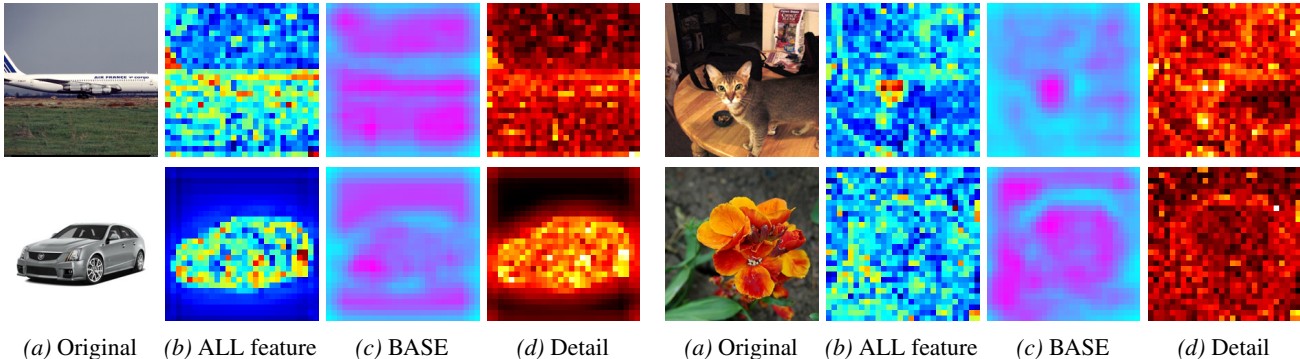

*(a) Original*  *(b) ALL feature*  *(c) BASE*  *(d) Detail*  *(a) Original*  *(b) ALL feature*  *(c) BASE*  *(d) Detail*

*Figure 3.* Visualization of SpecPL's spatial–spectral proxy decomposition. For each example, we show the original image, the full representation (ALL), and the disentangled Base (low-frequency proxy) and Detail (high-frequency proxy) components. The Base captures stable semantic invariants and global structure (e.g., shape/topology), while the Detail highlights fine-grained textures and local variations that provide discriminative evidence, helping alleviate the stability–generalization tension in prompt learning.

## 3. SpecPL: Disentangling Spectral Granularity for Prompt Learning

**Overview.** As illustrated in Fig. 2, SpecPL tackles modality asymmetry in prompt learning for VLMs by explicitly modeling the correspondence between *visual granularity* and *textual semantics*. Given an image $\mathbf{x}$, we introduce a *Spatial-Spectral Proxy* in the spatial VAE latent space of a frozen teacher to disentangle a low-frequency band capturing semantic invariants and a high-frequency band capturing instance-specific details. We project both bands into the CLIP embedding space with lightweight heads $\mathrm{Proj}_{\mathrm{low/high}}$, producing $\mathbf{t}^{\mathrm{low}}$ and $\mathbf{t}^{\mathrm{high}}$. These lightweight heads are learned jointly with the prompt parameters while keeping the CLIP backbone frozen, and we detach the VAE latents to prevent gradients from flowing into the teacher. We then initialize a *Visual Semantic Bank* $\mathbf{B}$ from low-band prototypes and perform *compositional text refinement* via soft retrieval. During **training only**, we further introduce factual and counterfactual granule supervision through a FiLM-style modulation branch to enforce sensitivity to fine-grained evidence. At **inference**, SpecPL requires neither the VAE teacher nor FiLM modulation; it relies solely on bank-based text refinement and standard CLIP similarity.

### 3.1. Preliminary: Multi-modal Prompt Learning

Contrastive VLMs such as CLIP align images and texts in a shared $d$-dimensional embedding space. Let $f_\theta$ and $g_\phi$ denote frozen visual and textual encoders, respectively. In multi-modal prompting (e.g., MaPLe (Khattak et al., 2023a)), prompts are inserted into both encoders:

$$\mathbf{v} = f_\theta(\mathbf{x}, \mathbf{P}_v), \qquad \mathbf{t} = g_\phi(\mathbf{y}, \mathbf{P}_t), \qquad (1)$$

where $\mathbf{v}, \mathbf{t} \in \mathbb{R}^d$ are $\ell_2$-normalized image/text embeddings and $\mathbf{P}_v = \varnothing$ if unused. Existing paradigms largely treat $\mathbf{v}$ as holistic and do not explicitly expose a granularity hierarchy,

limiting fine-grained and robust generalization.

### 3.2. Latent Granularity Factorization

Instead of pixel-space frequency analysis, we decompose granularity in the topology-preserving latent manifold of a pretrained, frozen VAE. Let $\mathbf{z} = \mathcal{E}(\mathbf{x}) \in \mathbb{R}^{C \times h \times w}$ denote the latent tensor, which preserves spatial correspondence to the input.

**Spatial-Spectral Proxy (low-pass + residual high-pass).** We define a computationally efficient proxy to separate stable base structure from detailed variations by local smoothing:

$$\mathbf{z}^{\mathrm{base}} = \mathcal{S}_k(\mathbf{z}), \qquad \mathbf{z}^{\mathrm{detail}} = \mathbf{z} - \mathbf{z}^{\mathrm{base}}, \qquad (2)$$

where $\mathcal{S}_k(\cdot)$ is a stride-1 $k \times k$ averaging operator without downsampling. $\mathbf{z}^{\mathrm{base}}$ suppresses local spatial fluctuations and captures stable structure (semantic invariants), while $\mathbf{z}^{\mathrm{detail}}$ isolates residual granules such as texture and instance-specific cues.

**Projection to CLIP space.** We extract spatial statistics $\phi(\cdot)$ and project each band via lightweight MLP heads:

$$\mathbf{t}^{\mathrm{low}} = \mathrm{Proj}_{\mathrm{low}}\big(\phi(\mathbf{z}^{\mathrm{base}})\big), \qquad \mathbf{t}^{\mathrm{high}} = \mathrm{Proj}_{\mathrm{high}}\big(\phi(\mathbf{z}^{\mathrm{detail}})\big), \qquad (3)$$

where $\phi(\mathbf{z}) = \frac{1}{hw}\sum_{i,j}|\mathbf{z}_{:,i,j}|$ computes mean absolute channel activation, and $\mathbf{t}^{\mathrm{low}}, \mathbf{t}^{\mathrm{high}} \in \mathbb{R}^d$ are $\ell_2$-normalized. The VAE teacher is frozen and provides these band features during training and, when used, bank initialization.

### 3.3. Visual Semantic Bank and Compositional Text Refinement

To bridge the modality gap and improve generalization to unseen categories, we maintain a frozen *Visual Semantic Bank* $\mathbf{B} = \{\mathbf{b}_m\}_{m=1}^M \in \mathbb{R}^{M \times d}$, where each entry represents a low-band visual primitive.

**Bank initialization.** We initialize $\mathbf{B}$ from low-band prototypes $\mathbf{t}^{\text{low}}$ sampled from training images (e.g., a few initial batches). We sequentially fill bank slots with $\mathbf{t}^{\text{low}}$; once full, each new $\mathbf{t}^{\text{low}}$ performs a nearest-neighbor assignment and updates its closest entry by an EMA rule:

$$m^* = \arg\max_m \langle \mathbf{t}^{\text{low}}, \mathbf{b}_m \rangle,$$

$$\mathbf{b}_{m^*} \leftarrow (1 - \mu)\mathbf{b}_{m^*} + \mu\,\mathbf{t}^{\text{low}}. \tag{4}$$

where $\mu \in (0, 1)$ is a momentum coefficient. The update is performed *outside* backpropagation; $\mathbf{B}$ is stored as frozen parameters and receives no gradients.

**Optional refresh.** During training, we may periodically refresh $\mathbf{B}$ with the same nearest-neighbor EMA update using incoming $\mathbf{t}^{\text{low}}$. This improves coverage of base-band primitives without introducing additional trainable parameters. In all cases, $\mathbf{B}$ remains frozen from the optimization standpoint with non-differentiable updates only.

**Compositional retrieval.** Given a class-specific text feature $\mathbf{t}_c$, we retrieve a compositional visual context by soft attention:

$$\alpha_{c,m} = \text{softmax}_m\big(\langle \mathbf{t}_c, \mathbf{b}_m \rangle / \tau\big), \qquad \mathbf{r}_c = \sum_{m=1}^{M} \alpha_{c,m} \mathbf{b}_m, \tag{5}$$

where $\tau$ is the temperature. $\mathbf{r}_c$ composes class appearance from reusable base-band primitives.

**Text refinement.** We fuse the original text feature with retrieved context via a residual aggregator:

$$\tilde{\mathbf{t}}_c = \text{LN}\Big(\mathbf{t}_c + \text{Agg}([\mathbf{t}_c; \mathbf{r}_c])\Big), \tag{6}$$

where $\text{Agg}(\cdot)$ is a lightweight MLP and LN is layer normalization. The refined feature $\tilde{\mathbf{t}}_c$ anchors prompts to stable base-band visual evidence.

### 3.4. Granule Modulation via Shared-Individual Fusion

While $\tilde{\mathbf{t}}_c$ enhances class semantics, we introduce a **training-only** granule modulation branch for detail-band sensitivity. We form a shared anchor $\mathbf{s}_i$ from either the label-indexed class text feature (raw/refined) or the image embedding $\mathbf{v}_i$ (Sec. 3.1); the label-indexed option is used only in this auxiliary branch. We denote the fusion as $F(\cdot, \cdot)$ and fuse $\mathbf{s}_i$ with the instance granule $\mathbf{t}_i^{\text{high}}$:

$$\mathbf{c}_i = F(\mathbf{s}_i, \mathbf{t}_i^{\text{high}}) := \text{LN}\Big(\mathbf{s}_i + \text{MLP}_{\text{fuse}}([\mathbf{s}_i; \mathbf{t}_i^{\text{high}}])\Big). \tag{7}$$

**FiLM granule modulator.**

$$[\gamma_i, \beta_i] = \text{MLP}_{\text{mod}}(\mathbf{c}_i),$$

$$\mathbf{v}_i^g \triangleq \text{FiLM}(\mathbf{c}_i, \mathbf{v}_i)$$

$$= \text{Norm}\Big((1 + \tanh(\gamma_i)) \odot \mathbf{v}_i + \beta_i\Big). \tag{8}$$

**Used only for auxiliary granule losses during training (not main logits nor inference).**

### 3.5. Training Objective

Our total objective combines the standard classification loss (using refined text) with three auxiliary terms:

$$\mathcal{L} = \mathcal{L}_{\text{cls}} + \lambda_1 \mathcal{L}_{\text{sem}} + \lambda_2 \mathcal{L}_g^f + \lambda_3 \mathcal{L}_g^{cf}. \tag{9}$$

**Main classification loss.** We compute logits using the frozen CLIP image embedding $\mathbf{v}_i$ and refined class text features $\tilde{\mathbf{T}} = \{\tilde{\mathbf{t}}_c\}_{c=1}^C$:

$$\mathcal{L}_{\text{cls}} = \text{CE}\Big(s \cdot \mathbf{v}_i \tilde{\mathbf{T}}^\top, y_i\Big), \tag{10}$$

where $s$ is the CLIP logit scale. This form matches inference.

**Label-free semantic alignment.** We compute a pseudo-label distribution $\mathbf{p}_i$ using stop-gradient on the current classifier output:

$$\mathbf{p}_i = \text{softmax}\Big(\text{stopgrad}\Big(s\,\mathbf{v}_i \tilde{\mathbf{T}}^\top\Big)\Big). \tag{11}$$

We then form the expected text direction using the learnable raw text features $\mathbf{T} = \{\mathbf{t}_c\}_{c=1}^C$:

$$\mathbf{t}_{\exp,i} = \sum_{c=1}^{C} p_{i,c}\,\text{Norm}(\mathbf{t}_c), \tag{12}$$

$$\mathcal{L}_{\text{sem}} = \mathbb{E}_i\big[1 - \cos\big(\mathbf{t}_{\exp,i}, \mathbf{t}_i^{\text{low}}\big)\big]. \tag{13}$$

where $\text{Norm}(\cdot)$ denotes $\ell_2$ normalization. Stop-gradient on $\mathbf{p}_i$ prevents pseudo-labels from backpropagating to the classifier. Gradients from $\mathcal{L}_{\text{sem}}$ update the prompt-induced text embeddings to align with the teacher low-band signal, jointly constrained by $\mathcal{L}_{cls}$.

**Factual granule supervision (train-only FiLM branch).** We supervise the FiLM-modulated embedding to encourage discriminative detail-band granules:

$$\mathcal{L}_g^f = \text{CE}\Big(s \cdot \mathbf{v}_i^g \mathbf{T}^\top, y_i\Big), \tag{14}$$

where $\mathbf{T}$ denotes the (raw) class text features.

**Counterfactual granule supervision.** We randomly permute batch indices by $\pi$ and swap detail-band granules:

$$\mathbf{c}_i^{cf} = F(\mathbf{s}_i, \mathbf{t}_{\pi(i)}^{\text{high}}), \qquad \mathbf{v}_i^{g,cf} = \text{FiLM}(\mathbf{c}_i^{cf}, \mathbf{v}_i), \tag{15}$$

and enforce the prediction to match the **granule source label** $y_{\pi(i)}$:

$$\mathcal{L}_g^{cf} = \text{CE}\Big(s \cdot \mathbf{v}_i^{g,cf} \mathbf{T}^\top, y_{\pi(i)}\Big). \tag{16}$$

This enforces granule sensitivity: if swapped granules change the predicted identity, the model must rely on fine-grained evidence rather than "shape-only" shortcuts.

## 3.6. Inference

At inference, SpecPL removes the frozen VAE teacher and the FiLM granule branch; we only refine text features with $\mathbf{B}$ and compute CLIP logits. We mix raw and refined text features (default $\eta = 1$):

$$\mathbf{t}_c^{\text{mix}} = (1 - \eta)\mathbf{t}_c + \eta\tilde{\mathbf{t}}_c, \tag{17}$$

where $\mathbf{t}_c$ and $\tilde{\mathbf{t}}_c$ are the raw and refined features, and $\mathbf{T}^{\text{mix}} = [\mathbf{t}_1^{\text{mix}}; \ldots; \mathbf{t}_C^{\text{mix}}]$.

$$\text{logits} = s \cdot \mathbf{v}\left(\mathbf{T}^{\text{mix}}\right)^\top. \tag{18}$$

# 4. Experiments

## 4.1. Experimental Setup

**Benchmarks and settings.** We evaluate base-to-novel generalization on 11 datasets (Deng et al., 2009; Fei-Fei et al., 2004; Parkhi et al., 2012; Krause et al., 2013; Nilsback & Zisserman, 2008; Bossard et al., 2014; Maji et al., 2013; Xiao et al., 2010; Cimpoi et al., 2014; Helber et al., 2019; Soomro et al., 2012) under the 16-shot protocol used in prior prompt-learning work (Zhou et al., 2022b;a; Khattak et al., 2023a; Li et al., 2025). We also report cross-dataset transfer (ImageNet → 10 targets) and domain generalization (ImageNet → V2/Sketch/A/R).

**Implementation details.** We use CLIP ViT-B/16 as in prior work (Zhou et al., 2022b;a; Khattak et al., 2023a) and follow each baseline's official training recipe (optimizer, LR schedule, epochs, batch size, augmentation). SpecPL adds a frozen VAE teacher for dual-band (low/high) guidance during training and a frozen semantic prototype bank for retrieval-based text refinement; the VAE teacher is removed at inference. All results are averaged over 3 seeds; hyperparameters are provided in the supplementary material. We use a base-only validation split (no novel data) and select checkpoints by base-class validation accuracy.

## 4.2. Base-to-Novel Generalization

**Main Results.** Table 1 shows that SpecPL improves HM across evaluated hosts, with larger gains on weaker baselines: CoOp/CoCoOp gain +4.86/+2.00 HM, while MaPLe/MMRL gain +0.80/+0.31 HM, consistent with complementary *visual-side* guidance beyond text-only prompt tuning.

**Generalization Gap.** We measure the *relative* base-to-novel gap in percent, $\mathcal{G}(\%) = 100 \times (\text{Base} - \text{Novel})/\text{Base}$ (lower is better). Fig. 4 reports mean $\mathcal{G}(\%)$ over 11 datasets: SpecPL reduces $\mathcal{G}$ by 31.6% for CoOp and 3.3% for MMRL, while leaving CoCoOp nearly unchanged, indicating that HM gains are mainly driven by improved transfer to novel classes.

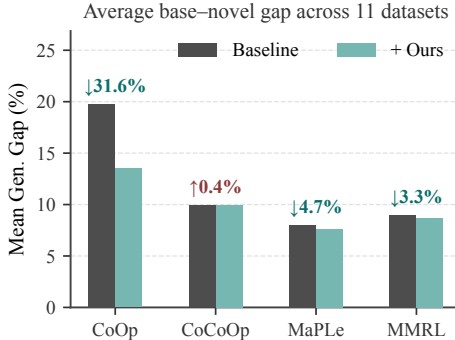

*Figure 4.* **Mean base–novel generalization gap (%).** $\mathcal{G}(\%) = 100 \times (\text{Base} - \text{Novel})/\text{Base}$ averaged over 11 datasets (lower is better). We also report relative gap reduction (%) w.r.t. each baseline.

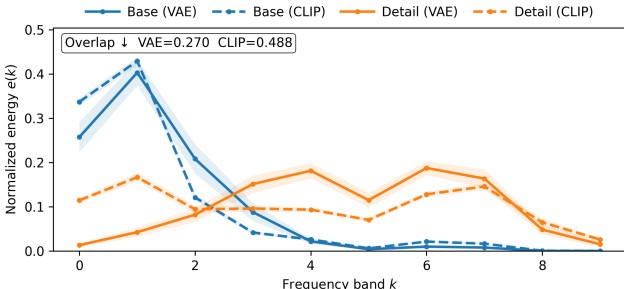

*Figure 5.* **Spectral diagnostic on ImageNet.** Normalized radial-band spectral energy $e(k)$ (mean±std over 200 images) for Base and Detail under Route-A decomposition ($k=7$, $K=10$, aligned to $14 \times 14$). Lower overlap indicates cleaner Base/Detail separation; the VAE shows markedly lower overlap than CLIP (0.270 vs. 0.488). Additional plots are in the appendix.

**Spectral Diagnostic of Base–Detail Separability.** We evaluate the suitability of a frozen VAE as a granularity teacher by measuring Base/Detail spectral separability under the same Route-A decomposition. We quantify mixing by spectral overlap (lower is better):

$$\text{Overlap} = \sum_{k=1}^{K} \min\left(e_{\text{base}}(k), e_{\text{detail}}(k)\right).$$

Fig. 5 shows substantially lower overlap for VAE than CLIP (0.270 vs. 0.488), supporting cleaner Base/Detail separation in the VAE manifold.

**Dataset-Specific Insights.** SpecPL yields larger gains on datasets where discriminative *texture* and fine-grained details matter. For instance, on **FGVC-Aircraft** and **DTD**, SpecPL boosts CoOp's HM from **28.75→35.98** and **54.24→64.06**, respectively (Table 1). We also observe improvements on shift-heavy targets such as **EuroSAT** (**68.69→73.83** HM), consistent with better robustness to style/domain changes. These results suggest that spectral granularity is most beneficial when class identity hinges on subtle appearance cues rather than coarse semantics.

| Method | Average | | | ImageNet | | | Caltech101 | | | OxfordPets | | |
|---|---|---|---|---|---|---|---|---|---|---|---|---|
| | Base | Novel | HM | Base | Novel | HM | Base | Novel | HM | Base | Novel | HM |
| CLIP (ICML 21) | 69.34 | 74.22 | 71.70 | 72.43 | 68.14 | 70.22 | 96.84 | 94.00 | 95.40 | 91.17 | 97.26 | 94.12 |
| CoOp (IJCV 22) | 82.69 | 63.22 | 71.66 | 76.47 | 67.88 | 71.92 | 98.00 | 89.81 | 93.73 | 93.67 | 95.29 | 94.47 |
| CoCoOp (CVPR 22) | 80.47 | 71.69 | 75.83 | 75.98 | 70.43 | 73.10 | 97.96 | 93.81 | 95.84 | 95.20 | 97.69 | 96.43 |
| MaPLe (CVPR 23) | 82.28 | 75.14 | 78.55 | 76.66 | 70.54 | 73.47 | 97.74 | 94.36 | 96.02 | 95.43 | 97.76 | 96.58 |
| PromptSRC (ICCV 23) | 84.26 | 76.10 | 79.97 | 77.60 | 70.73 | 74.01 | 98.10 | 94.03 | 96.02 | 95.33 | 97.30 | 96.30 |
| ArGue (CVPR 24) | 83.69 | 78.07 | 80.78 | 76.92 | 72.06 | 74.41 | 98.43 | 95.20 | 96.79 | 95.36 | 97.95 | 96.64 |
| DePT (CVPR 24) | 83.66 | 71.82 | 77.29 | 77.13 | 70.10 | 73.45 | 98.33 | 94.33 | 96.29 | 94.70 | 97.63 | 96.14 |
| CoPrompt (ICLR 24) | 84.00 | 77.23 | 80.48 | 77.67 | 71.27 | 74.33 | 98.27 | 94.90 | 96.55 | 95.67 | 98.10 | 96.87 |
| MMRL (CVPR 25) | 85.68 | 77.16 | 81.20 | 77.90 | 71.30 | 74.45 | 98.97 | 94.50 | 96.68 | 95.90 | 97.60 | 96.74 |
| CoOp + Ours | 83.32 | 70.74 | **76.52** (+4.86) | 77.01 | 70.01 | **73.34** | 98.26 | 92.54 | **95.31** | 95.04 | 96.55 | **95.79** |
| CoCoOp + Ours | 82.84 | 73.79 | **77.83** (+2.00) | 74.95 | 70.25 | 72.52 | 97.74 | 94.69 | **96.19** | 95.04 | 97.91 | **96.45** |
| MaPLe + Ours | 83.02 | 76.00 | **79.35** (+0.80) | 76.97 | 69.37 | 72.97 | 98.39 | 94.87 | **96.60** | 95.76 | 97.87 | **96.81** |
| MMRL + Ours | 85.90 | 77.55 | **81.51** (+0.31) | 78.05 | 71.50 | 74.63 | 99.10 | 94.43 | 96.71 | 96.20 | 97.40 | **96.80** |

| Method | StanfordCars | | | Flowers102 | | | Food101 | | | FGVCAircraft | | |
|---|---|---|---|---|---|---|---|---|---|---|---|---|
| | Base | Novel | HM | Base | Novel | HM | Base | Novel | HM | Base | Novel | HM |
| CLIP (ICML 21) | 63.37 | 74.89 | 68.65 | 72.08 | 77.80 | 74.83 | 90.10 | 91.22 | 90.66 | 27.19 | 36.29 | 31.09 |
| CoOp (IJCV 22) | 78.12 | 60.40 | 68.13 | 97.60 | 59.67 | 74.06 | 88.33 | 82.26 | 85.19 | 40.44 | 22.30 | 28.75 |
| CoCoOp (CVPR 22) | 70.49 | 73.59 | 72.01 | 94.87 | 71.75 | 81.71 | 90.70 | 91.29 | 90.99 | 33.41 | 23.71 | 27.74 |
| MaPLe (CVPR 23) | 72.94 | 74.00 | 73.47 | 95.92 | 72.46 | 82.56 | 90.71 | 92.05 | 91.38 | 37.44 | 35.61 | 36.50 |
| PromptSRC (ICCV 23) | 78.27 | 74.97 | 76.58 | 98.07 | 76.50 | 85.95 | 90.67 | 91.53 | 91.10 | 42.73 | 37.87 | 40.15 |
| ArGue (CVPR 24) | 75.64 | 73.38 | 74.49 | 98.34 | 75.41 | 85.36 | 92.33 | 91.96 | 92.14 | 40.46 | 38.03 | 39.21 |
| DePT (CVPR 24) | 79.67 | 72.40 | 75.86 | 98.20 | 72.00 | 83.08 | 90.43 | 91.33 | 90.88 | 42.53 | 22.53 | 29.46 |
| CoPrompt (ICLR 24) | 76.97 | 74.40 | 75.66 | 97.27 | 76.60 | 85.71 | 90.73 | 92.07 | 91.40 | 40.20 | 39.33 | 39.76 |
| MMRL (CVPR 25) | 81.30 | 75.07 | 78.06 | 98.97 | 77.27 | 86.78 | 90.57 | 91.50 | 91.03 | 46.30 | 37.03 | 41.15 |
| CoOp + Ours | 79.16 | 67.83 | **73.06** | 98.07 | 70.66 | **82.14** | 89.16 | 89.40 | **89.28** | 42.40 | 31.25 | **35.98** |
| CoCoOp + Ours | 78.50 | 69.13 | **73.52** | 97.53 | 70.43 | **81.80** | 89.53 | 90.97 | **90.24** | 42.30 | 34.07 | **37.74** |
| MaPLe + Ours | 75.89 | 73.14 | **74.49** | 97.12 | 73.71 | **83.81** | 90.58 | 91.80 | 91.18 | 37.85 | 37.43 | **37.64** |
| MMRL + Ours | 82.20 | 75.50 | **78.71** | 98.80 | 77.20 | 86.67 | 91.00 | 91.70 | **91.35** | 46.20 | 37.80 | **41.58** |

| Method | SUN397 | | | DTD | | | EuroSAT | | | UCF101 | | |
|---|---|---|---|---|---|---|---|---|---|---|---|---|
| | Base | Novel | HM | Base | Novel | HM | Base | Novel | HM | Base | Novel | HM |
| CLIP (ICML 21) | 69.36 | 75.35 | 72.23 | 53.24 | 59.90 | 56.37 | 56.48 | 64.05 | 60.03 | 70.53 | 77.50 | 73.85 |
| CoOp (IJCV 22) | 80.60 | 65.89 | 72.51 | 79.44 | 41.18 | 54.24 | 92.19 | 54.74 | 68.69 | 84.69 | 56.05 | 67.46 |
| CoCoOp (CVPR 22) | 79.74 | 76.86 | 78.27 | 77.01 | 56.00 | 64.85 | 87.49 | 60.04 | 71.21 | 82.33 | 73.45 | 77.64 |
| MaPLe (CVPR 23) | 80.82 | 78.70 | 79.75 | 80.36 | 59.18 | 68.16 | 94.07 | 73.23 | 82.35 | 83.00 | 78.66 | 80.77 |
| PromptSRC (ICCV 23) | 82.67 | 78.47 | 80.52 | 83.37 | 62.97 | 71.75 | 92.90 | 73.90 | 82.32 | 87.10 | 78.80 | 82.74 |
| ArGue (CVPR 24) | 81.52 | 80.74 | 81.13 | 81.60 | 66.55 | 73.31 | 94.43 | 88.24 | 91.23 | 85.56 | 79.29 | 82.31 |
| DePT (CVPR 24) | 82.37 | 75.07 | 78.55 | 83.20 | 56.13 | 67.04 | 88.27 | 66.27 | 75.70 | 85.43 | 72.17 | 78.24 |
| CoPrompt (ICLR 24) | 82.63 | 80.03 | 81.30 | 83.13 | 64.73 | 72.79 | 94.60 | 78.57 | 85.84 | 86.90 | 79.57 | 83.07 |
| MMRL (CVPR 25) | 83.20 | 79.30 | 81.20 | 85.67 | 65.00 | 73.82 | 95.60 | 80.17 | 87.21 | 88.10 | 80.07 | 83.89 |
| CoOp + Ours | 81.09 | 70.06 | **75.17** | 80.83 | 53.06 | **64.06** | 89.96 | 62.60 | **73.83** | 85.14 | 67.06 | **75.03** |
| CoCoOp + Ours | 77.75 | 77.85 | 77.80 | 81.80 | 55.07 | **65.82** | 90.67 | 75.00 | **82.09** | 85.43 | 75.37 | **80.08** |
| MaPLe + Ours | 81.53 | 77.14 | 79.28 | 81.83 | 60.95 | 69.86 | 92.66 | 80.44 | **86.12** | 84.63 | 79.25 | **81.85** |
| MMRL + Ours | 83.57 | 79.11 | **81.28** | 85.52 | 65.72 | **74.32** | 95.40 | 81.45 | **87.87** | 88.85 | 81.20 | **84.85** |

*Table 1.* Base-to-novel generalization on 11 datasets (16-shot). The upper part lists representative and recent prompt-learning methods. The lower part evaluates SpecPL as a plug-in booster on top of representative baselines (+Ours).

| Method | Source | Target Dataset | | | | | | | | | | Average |
|---|---|---|---|---|---|---|---|---|---|---|---|---|
| | Image Net | Caltech 101 | Oxford Pets | Stanford Cars | Flowers 102 | Food101 | FGVC Aircraft | SUN397 | DTD | Euro SAT | UCF101 | |
| CoOp | 71.51 | 93.70 | 89.14 | 64.51 | 68.71 | 85.30 | 18.47 | 64.15 | 41.92 | 46.39 | 66.55 | 63.88 |
| +Ours | 72.12 | 94.12 | 90.62 | 63.66 | 71.13 | 86.09 | 23.85 | 65.71 | 45.04 | 53.54 | 67.38 | **66.11** (+2.23) |
| MMRL | 72.03 | 94.67 | 91.43 | 66.10 | 72.77 | 86.40 | 26.30 | 67.57 | 45.90 | 53.10 | 68.27 | 67.25 |
| +Ours | 72.07 | 94.85 | 91.99 | 66.48 | 72.96 | 86.86 | 26.58 | 67.94 | 46.43 | 52.88 | 69.10 | **67.61** (+0.36) |

*Table 2.* Cross-dataset transfer (ImageNet $\to$ 10 target datasets). SpecPL improves the average transfer accuracy on representative baselines.

| Method | Source | Target Dataset | | | | Average |
|---|---|---|---|---|---|---|
| | ImageNet | -V2 | -S | -A | -R | |
| CoOp | 71.51 | 64.20 | 47.99 | 49.71 | 75.21 | 59.28 |
| +Ours | 72.12 | 65.25 | 48.25 | 50.45 | 76.30 | **60.06** (+0.78) |
| MMRL | 72.03 | 64.47 | 49.17 | 51.20 | 77.53 | 60.59 |
| +Ours | 72.07 | 64.60 | 49.33 | 51.43 | 77.67 | **60.76** (+0.17) |

*Table 3.* Domain generalization on ImageNet shifts (V2/Sketch/A/R). SpecPL improves average robustness on representative baselines.

| Variant | Bank | Sem | $G_F$ | $G_{CF}$ | Base | Novel | HM | $\Delta$HM |
|---|---|---|---|---|---|---|---|---|
| CoOp | | | | | 40.44 | 22.30 | 28.75 | -7.23 |
| Raw+$G_{CF}$ | | | | ✓ | 41.17 | 17.10 | 24.16 | -11.82 |
| Raw+$G_F$ | | | ✓ | | 40.27 | 18.13 | 25.01 | -10.97 |
| Raw+$G_F$+$G_{CF}$ | | | ✓ | ✓ | 41.33 | 17.60 | 24.69 | -11.29 |
| Bank | ✓ | | | | 43.03 | 19.60 | 26.93 | -9.05 |
| +Sem | ✓ | ✓ | | | 43.93 | 20.50 | 27.96 | -8.02 |
| +$G_F$ | ✓ | | ✓ | | 41.75 | 20.25 | 27.27 | -8.71 |
| +$G_{CF}$ | ✓ | | | ✓ | 44.09 | 23.09 | 30.31 | -5.67 |
| +$G_F$+$G_{CF}$ | ✓ | | ✓ | ✓ | 43.73 | 22.38 | 29.60 | -6.38 |
| +Sem+$G_F$ | ✓ | ✓ | ✓ | | 42.50 | 30.65 | 35.62 | -0.36 |
| +Sem+$G_{CF}$ | ✓ | ✓ | | ✓ | 43.46 | 27.77 | 33.89 | -2.09 |
| +Ours | ✓ | ✓ | ✓ | ✓ | 42.40 | 31.25 | 35.98 | +0.00 |

*Table 4.* **Ablation on CoOp / FGVC-Aircraft** (base-to-novel, 16-shot). Rows with Bank enabled use refined text; "Raw+" rows use raw CoOp text prompts without bank refinement. $\Delta$HM is computed w.r.t. CoOp+Ours (HM=35.98).

## 4.3. Cross-dataset Evaluation

**Results.** Table 2 shows that SpecPL improves cross-dataset transfer on both baselines (+2.23 on CoOp; +0.36 on MMRL on average). Gains are stronger on texture/fine-grained or shift-heavy targets (e.g., FGVC-Aircraft/DTD/EuroSAT), with small changes on the remaining datasets.

## 4.4. Domain Generalization

**Results.** Table 3 shows that SpecPL improves average robustness under natural, adversarial, and stylistic shifts on both baselines (+0.78 on CoOp and +0.17 on MMRL). While the gains are modest on the strong MMRL baseline, they are consistent across all target domains, suggesting that SpecPL complements existing prompt learning designs for handling domain shifts without additional inference-time overhead.

## 4.5. Ablation Study

We conduct ablations under base-to-novel (16-shot) on FGVC-Aircraft with CoOp as baseline (Table 4).

**Visual Semantic Bank provides an anchor but is not sufficient.** Operating on raw text without the bank ("Raw+" rows) causes large drops (HM ≈ 24–25%), below the CoOp baseline (28.75%), showing that retrieval-based refinement stabilizes adaptation. The bank alone is conservative: it

increases Base accuracy (43.03 vs. 40.44) but suppresses Novel performance (19.60 vs. 22.30), yielding a lower HM (26.93%). Thus, the bank provides a stable anchor but benefits from training guidance on fine-grained novel classes.

**Semantic alignment stabilizes factual granule modulation.** On top of the bank, adding semantic alignment alone (+Sem, HM 27.96%) or factual granule modulation alone (+$G_F$, HM 27.27%) does not outperform CoOp. In contrast, combining them (+Sem+$G_F$) yields a substantial improvement to 35.62% HM. This indicates that semantic alignment provides a stable structural basis in the latent space, enabling factual granule modulation to inject discriminative details rather than introducing noisy perturbations.

**Counterfactual supervision complements the full model under a stable backbone.** Without semantic alignment, combining $G_F$ and $G_{CF}$ can interfere (+$G_F$+$G_{CF}$, HM 29.60% < +$G_{CF}$, HM 30.31%), suggesting that counterfactual swapping may introduce ambiguous supervision when semantics are not well anchored. When built upon the strong +Sem+$G_F$ backbone, adding $G_{CF}$ provides a modest but consistent gain (+0.36 HM), reaching the best performance (35.98% HM). This supports that counterfactual granule supervision is most effective when grounded on stable semantics and factual granularity.

**Teacher & Disentanglement.** As detailed in Appendix Tables 7 and 8, while both explicit disentanglement and the VAE teacher independently improve generalization, the VAE representation space proves more effective, and their combination achieves the optimal base-to-novel transfer.

## 5. Conclusion

In this work, we identify the lack of explicit visual granularity factorization as a key bottleneck behind modality asymmetry in VLM adaptation. To bridge this gap, we propose **SpecPL**, which reformulates prompt learning with a lightweight *spatial–spectral proxy*. Unlike pixel-level frequency analysis, SpecPL decomposes visual signals directly in the spatially aligned latent manifold of a frozen VAE, disentangling semantic invariants from fine-grained details. This enables a complementary mechanism: a *Visual Semantic Bank* anchors prompts to stable low-frequency semantic **primitives**, while *counterfactual granule supervision* intervenes on the high-frequency component with invariants fixed, encouraging sensitivity to discriminative granularity. Extensive experiments show that SpecPL consistently improves harmonic-mean (HM) accuracy on base-to-novel generalization and cross-dataset transfer, with particularly strong gains on fine-grained domains. These results highlight a promising direction for VLM adaptation: moving beyond text-centric optimization to explicitly model structured visual granularity in deep representations.

## Impact Statement

This paper studies parameter-efficient adaptation of vision–language models (VLMs) by explicitly modeling visual granularity during training and refining textual prompts at inference. Our goal is to improve robustness and generalization of VLMs under distribution shift and fine-grained recognition, while keeping inference lightweight.

**Potential positive impacts.** More reliable VLM adaptation can reduce the need for task-specific full fine-tuning and enable broader deployment in resource-constrained settings. Improved robustness to domain shift may benefit applications such as visual inspection, assistive interfaces, and content understanding, where distribution mismatches commonly occur. In addition, our granularity-aware formulation offers a more interpretable handle for analyzing when prompt learning succeeds or fails (e.g., coarse- vs. fine-grained categories).

**Potential negative impacts and risks.** As with most improvements to generic VLM adaptation, the method could also be used to enhance downstream systems with harmful uses (e.g., surveillance or sensitive content analysis). Moreover, our training procedure relies on an external VAE teacher and large-scale VLMs; increased training complexity and compute may raise environmental and accessibility concerns. Finally, stronger adaptation under distribution shift does not remove dataset biases: models may still inherit spurious correlations or representational biases present in pretraining data.

**Mitigations and responsible use.** We do not collect new data and evaluate only on standard public benchmarks. We encourage reporting compute and energy usage when reproducing results, and we recommend auditing downstream deployments for dataset bias and misuse risks, especially in high-stakes or privacy-sensitive settings.

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

# A. Ethics and Reproducibility Statement

**Content Integrity.** We affirm that Large Language Models (LLMs) were utilized exclusively for linguistic refinement and grammatical polishing; all technical frameworks (including the Spatial-Spectral Proxy and Counterfactual Supervision), experimental analyses, and theoretical conclusions remain the original work of the authors.

**Data Compliance.** The empirical results presented in this study were derived from eleven established public benchmarks (e.g., ImageNet, EuroSAT) under their respective licenses and standard protocols. We do not collect any new data nor involve human subjects. Since these benchmarks were curated by third parties, they may contain societal biases or other unintended artifacts inherited from their collection processes. We therefore avoid making claims about the absence of bias. Regarding privacy, we do not use datasets that are intended to contain personally identifiable information (PII), and our method does not introduce any additional PII. Our counterfactual granules are synthetically generated within a frozen VAE latent space through stochastic permutations, requiring no additional real-world data collection.

**Reproducibility Commitment.** To ensure research transparency and reproducibility, we commit to releasing our full implementation, including the source code, training scripts, and the pre-computed Visual Semantic Bank, upon the paper's acceptance. Detailed hyperparameter configurations and computational costs are further provided in the supplementary material to facilitate exact replication of our results.

# B. Limitations

## B.1. Dependency on External VAE Teacher and Training Overhead.

SpecPL relies on a pre-trained, frozen VAE (e.g., Stable Diffusion's VAE) to provide the spatial-spectral proxy during training. While this external dependency is discarded during inference—ensuring zero additional FLOPs or latency overhead at test time—it inevitably introduces a minor training cost, primarily a one-time feature caching step and an increased memory footprint compared to standard prompt tuning methods. Although we show that the VAE manifold provides superior spectral disentanglement compared to CLIP's native features, future work may explore distillation to further minimize the training footprint, or tackle the open challenge of learning an intrinsic spectral decomposition directly within the CLIP encoder.

## B.2. Performance Boundary on Coarse-grained Tasks.

Our experiments indicate that SpecPL provides the most significant gains on datasets requiring fine-grained discrimination (e.g., FGVC-Aircraft, DTD) or handling distribution shifts. On generic object recognition benchmarks dominated by distinct semantic categories (e.g., standard ImageNet), the improvement over strong baselines is more marginal. SpecPL's core mechanism is designed to capture fine-grained high-frequency details; consequently, it yields diminishing returns when global shape or semantics are already sufficient for classification.

## B.3. Sensitivity to High-Frequency Domain Shifts.

Because SpecPL explicitly disentangles and relies on high-frequency signals for fine-grained discrimination, its effectiveness is intrinsically tied to the quality of these details. If applied to out-of-distribution data with severe high-frequency corruption (e.g., extreme JPEG artifacts, sensor noise, or adversarial high-frequency perturbations not present in standard benchmarks), the model may misconstrue this noise as discriminative evidence.

## B.4. Optimization Sensitivity of Counterfactual Supervision.

As analyzed in the ablation study, the Counterfactual Granule Supervision ($\mathcal{L}_g^{cf}$) acts as a strong regularizer. We observed that applying this objective aggressively without sufficient semantic anchoring (via the Visual Semantic Bank) can lead to training instability or suboptimal convergence. Achieving the optimal synergy requires a balanced weighting between the factual and counterfactual branches, which may necessitate hyperparameter tuning when transferring to significantly different domains.

# C. Future Work

### C.1. Broader VLM Applications in Detail-Sensitive Domains.

An important bottleneck in current VLM adaptation lies not only in alignment strength but also in the type of visual evidence being aligned. Since SpecPL highlights how granularity-aware adaptation can prevent text-only prompts from overfitting coarse semantics, a promising direction is extending this framework to domains where visual details are decisive rather than supplementary. Exploring how the properties of the teacher space can be utilized to absorb fine-grained visual information in fields like medical imaging diagnostics (Zhou et al., 2026),video understanding (Huang et al., 2024)and embodied agents (Zheng et al., 2025b) remains highly relevant.

### C.2. Extension to Spatiotemporal Domains.

Currently, SpecPL operates on static images. However, the concept of "spectral granularity" naturally extends to the temporal dimension in video understanding. High-frequency temporal fluctuations often correspond to motion dynamics, while low-frequency components capture static scenes. Extending the Spatial-Spectral Proxy to decompose video tokens could improve action recognition tasks where motion texture is the key discriminator.

### C.3. Adaptive Spectral Filtering.

In this work, we use a fixed pooling operation to separate Base and Detail components. Future work could explore *learnable* spectral filters or dynamic frequency gating mechanisms that allow the model to adaptively determine the optimal cut-off frequency for different images, potentially handling a wider range of visual granularities.

### C.4. Generative Applications.

While SpecPL focuses on discriminative tasks, the disentangled prompt representations (Base for structure, Detail for texture) hold promise for controllable text-to-image generation. Investigating how these disentangled prompts can guide diffusion models to generate consistent subjects with variable textures is an avenue for future exploration.

# D. Implementation Details

### D.1. Datasets

We evaluate SpecPL on **15** recognition benchmarks under three standard prompt-learning settings: (i) *base-to-novel generalization*, (ii) *cross-dataset transfer*, and (iii) *domain generalization*. For base-to-novel and cross-dataset transfer, we consider **11** diverse classification datasets: ImageNet-1K (Deng et al., 2009) and Caltech-101 (Fei-Fei et al., 2004) for generic object recognition; OxfordPets (Parkhi et al., 2012), StanfordCars (Krause et al., 2013), Flowers-102 (Nilsback & Zisserman, 2008), Food-101 (Bossard et al., 2014), and FGVC-Aircraft (Maji et al., 2013) for fine-grained recognition; SUN-397 (Xiao et al., 2010) for scene recognition; UCF-101 (Soomro et al., 2012) for action recognition; DTD (Cimpoi et al., 2014) for texture recognition; and EuroSAT (Helber et al., 2019) for satellite imagery recognition. For **cross-dataset transfer**, we train prompts on ImageNet-1K and evaluate on the remaining 10 datasets. For **domain generalization**, we use ImageNet-1K (Deng et al., 2009) as the source domain and evaluate on four distribution-shifted variants: ImageNet-V2 (Recht et al., 2019), ImageNet-Sketch (Wang et al., 2019), ImageNet-A (Hendrycks et al., 2021b), and ImageNet-R (Hendrycks et al., 2021a). Overall, this results in 15 evaluation datasets.

### D.2. Training and Reproducibility Setup

We implement SpecPL by extending standard prompt-learning baselines (e.g., CoOp/CoCoOp/Maple/MMRL). **Unless explicitly stated**, we strictly follow the **original training protocols of the corresponding baseline** for each setting, including the optimizer, learning-rate schedule, batch size, number of training epochs, data preprocessing/augmentation, and evaluation procedure. In all experiments, we keep the pre-trained vision–language backbone **frozen** and only optimize the prompt-related parameters together with the lightweight modules introduced by SpecPL, ensuring a fair and directly comparable evaluation. For each benchmark, we report the average performance over three random seeds.

| Parameter (as in figures) | Value | Description |
|---|---|---|
| **BANK Size** | 64 | Number of entries in the visual semantic/attribute bank. |
| **Visual Semantic Bank temperature ($\tau$)** | 0.07 | Softmax temperature $\tau$ controlling bank matching sharpness. |
| $\lambda_{\text{sem}}$ | 0.1 | Weight of semantic anchoring loss. |
| $\lambda_f$ | 0.1 | Weight of granule supervision for the factual branch ($g_f$). |
| $\lambda_{cf}$ | 0.1 | Weight of granule supervision for the counterfactual branch ($g_{cf}$). |
| **Input Modality** | `raw text` | Shared source modality: `raw text`/`refined text`/`image`. |
| $K_{\text{high}}$ | 7 | High-pass kernel size in the teacher/VAE. |
| $K_{\text{low}}$ | 7 | Low-pass kernel size. |
| **Teacher/VAE pretrained ID** | `REPA-E/e2e-qwenimage-vae` | Identifier of the frozen teacher/VAE checkpoint (default). |
| **Use bank refine** | True | Enable bank refinement. |

*Table 5.* Hyperparameters of **SpecPL** used in our experiments.

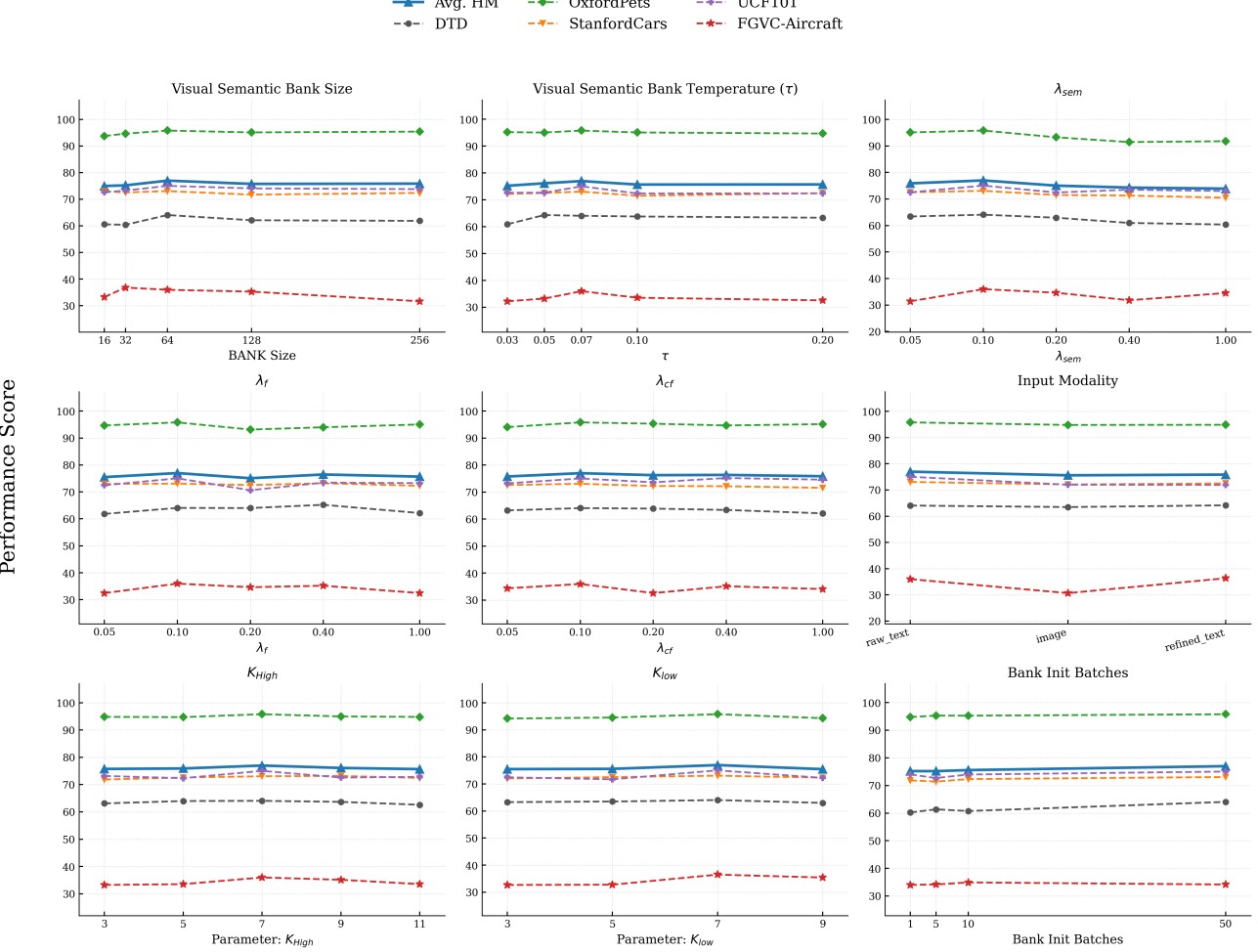

*Figure 6.* **Hyperparameter sensitivity across five datasets.** We report Base, Novel, and HM. The ablations quantify the contribution of each SpecPL component, and the sensitivity plots vary key hyperparameters (e.g., bank size, bank temperature, and shared input modality).

### D.3. Hyperparameters and Sensitivity

Table 5 summarizes the hyperparameters of SpecPL. Unless explicitly mentioned, we use a **single shared configuration across datasets** to avoid per-dataset tuning. In particular, we fix all loss weights to 0.1, which we found to provide stable trade-offs between base-class fitting and novel-class generalization. We keep the teacher/VAE configuration at its default settings and do not tune it.

Figure 6 reports both **component ablations** and **hyperparameter sensitivity** under the base-to-novel protocol (16-shot) on FGVC-Aircraft. Overall, the results support two key takeaways: (i) each component in SpecPL provides complementary gains, especially in improving novel-class performance and the harmonic mean; (ii) SpecPL is relatively robust to moderate changes of bank-related hyperparameters, with best performance typically achieved at intermediate bank sizes and temperatures, while overly small/large settings tend to hurt novel-class generalization.

### D.4. Architectural generalizability beyond CLIP

To examine whether the proposed spectral-granularity guidance is tied to CLIP-specific representations, we further evaluate SpecPL on a structurally different VLM backbone, BLIP-ITC (Li et al., 2022). We use a ViT-B/16 image encoder, a BERT-base text encoder, and the 256-dimensional image-text contrastive embedding space, and follow the CoOp-style prompt-learning protocol with 16 learnable context tokens. Compared with CLIP, this setting differs in the text encoder, projection dimensionality, and pre-training objectives, and therefore provides an additional test of backbone transferability.

| Dataset | Method | Base | Novel | HM |
|---|---|---|---|---|
| Caltech101 | BLIP-ITC | $86.42 \pm 0.16$ | $48.44 \pm 2.59$ | $62.05 \pm 2.14$ |
| | + SpecPL (Ours) | $\mathbf{87.54} \pm 0.78$ | $\mathbf{52.15} \pm 4.59$ | $\mathbf{65.29} \pm 3.65$ |
| DTD | BLIP-ITC | $\mathbf{74.04} \pm 0.75$ | $30.11 \pm 2.90$ | $42.75 \pm 2.91$ |
| | + SpecPL (Ours) | $73.50 \pm 0.42$ | $\mathbf{45.25} \pm 0.80$ | $\mathbf{56.01} \pm 0.73$ |
| EuroSAT | BLIP-ITC | $\mathbf{86.88} \pm 0.67$ | $45.94 \pm 5.50$ | $59.96 \pm 4.72$ |
| | + SpecPL (Ours) | $86.49 \pm 0.87$ | $\mathbf{52.05} \pm 4.93$ | $\mathbf{64.89} \pm 3.61$ |
| FGVCAircraft | BLIP-ITC | $19.21 \pm 0.03$ | $4.36 \pm 0.07$ | $7.11 \pm 0.09$ |
| | + SpecPL (Ours) | $\mathbf{21.21} \pm 0.90$ | $\mathbf{6.11} \pm 0.12$ | $\mathbf{9.49} \pm 0.20$ |
| OxfordPets | BLIP-ITC | $66.68 \pm 0.68$ | $43.61 \pm 5.65$ | $52.56 \pm 4.05$ |
| | + SpecPL (Ours) | $\mathbf{68.00} \pm 0.24$ | $\mathbf{60.10} \pm 2.19$ | $\mathbf{63.79} \pm 1.14$ |
| StanfordCars | BLIP-ITC | $29.73 \pm 0.48$ | $\mathbf{13.25} \pm 0.69$ | $18.31 \pm 0.58$ |
| | + SpecPL (Ours) | $\mathbf{34.13} \pm 0.56$ | $12.72 \pm 0.33$ | $\mathbf{18.53} \pm 0.42$ |
| UCF101 | BLIP-ITC | $61.98 \pm 0.29$ | $\mathbf{36.09} \pm 1.90$ | $45.60 \pm 1.47$ |
| | + SpecPL (Ours) | $\mathbf{64.50} \pm 0.75$ | $35.97 \pm 0.95$ | $\mathbf{46.17} \pm 0.78$ |
| **Average** | BLIP-ITC | 60.71 | 31.68 | 41.19 |
| | + SpecPL (Ours) | **62.19** | **37.76** | **46.31** |

*Table 6.* **Backbone transfer to BLIP-ITC.** We compare BLIP-ITC and BLIP-ITC + SpecPL under the base-to-novel protocol on seven benchmarks. Best results are highlighted in bold.

As shown in Table 6, SpecPL improves the average HM from 41.19 to 46.31, corresponding to a +5.12 point gain over the BLIP-ITC prompt-learning baseline. The improvement is primarily driven by novel-class generalization: average Novel accuracy increases from 31.68 to 37.76 (+6.08), while Base accuracy also improves from 60.71 to 62.19 (+1.48). The largest HM gains appear on datasets where local texture or fine-grained visual evidence is especially important, including DTD (+13.26), OxfordPets (+11.23), and EuroSAT (+4.93). These results mirror the trend observed with CLIP-based backbones: SpecPL improves transfer to novel categories without relying on backbone-specific architectural assumptions. The consistent gains across two structurally distinct VLM families suggest that spectral-granularity disentanglement captures a generalizable adaptation principle, rather than a heuristic tailored to CLIP.

### D.5. Additional Diagnostics

**Convergence of the auxiliary objectives.** Figure 7 verifies that the proposed auxiliary losses do not destabilize optimization. On DTD with CoOp + SpecPL, the average total loss over the last five epochs decreases from 2.0132 at the beginning of training to 0.6200 at convergence. The main classification loss decreases from 1.2907 to 0.0769, while the semantic-alignment and factual-granule losses decrease from 1.3643 to 0.7703 and from 0.9884 to 0.9019, respectively.

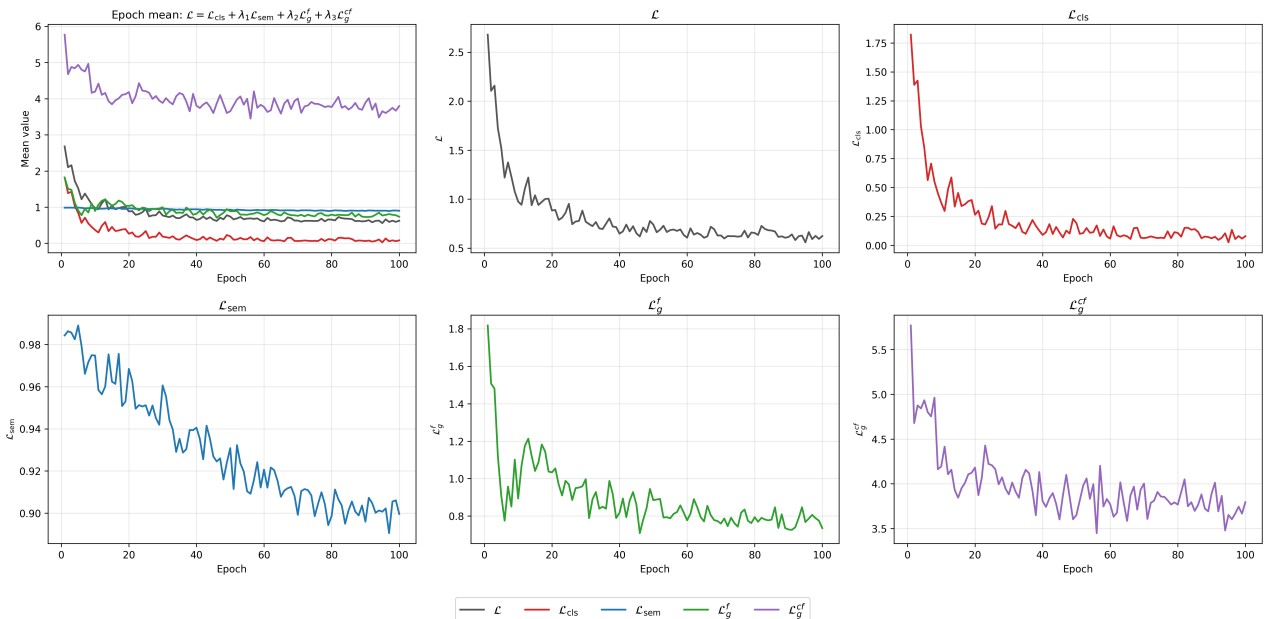

*Figure 7.* **Training convergence on DTD.** We plot epoch-wise optimization curves for CoOp + SpecPL under the 16-shot base-to-novel setting. The leftmost panel summarizes the total objective $\mathcal{L} = \mathcal{L}_{\text{cls}} + \lambda_1 \mathcal{L}_{\text{sem}} + \lambda_2 \mathcal{L}_g^f + \lambda_3 \mathcal{L}_g^{cf}$, and the remaining panels show the individual loss terms. The counterfactual granule loss is shown before weighting; its contribution to the total objective is scaled by $\lambda_3 = 0.1$.

| Method | Base | Novel | HM |
|---|---|---|---|
| CoOp | 82.69 | 63.22 | 71.66 |
| VAE + holistic | 83.45 | 68.48 | 75.23 |
| CLIP visual encoder (ViT) + disentangled | **83.48** | 68.12 | 75.02 |
| VAE + disentangled (SpecPL) | 83.32 | **70.74** | **76.52** |

*Table 7.* **Average teacher/decomposition ablation.** We report average Base, Novel, and HM results on the 11-dataset base-to-novel benchmark.

| Method | ImageNet | Caltech101 | OxfordPets | StanfordCars | Flowers102 | Food101 | FGVCAircraft | SUN397 | DTD | EuroSAT | UCF101 | Average |
|---|---|---|---|---|---|---|---|---|---|---|---|---|
| VAE + holistic HM | 72.97 | **96.14** | 94.98 | 71.02 | 79.35 | 89.17 | 33.71 | **75.37** | 62.39 | 73.23 | 73.39 | 75.23 |
| CLIP encoder (ViT) + disentangled HM | 72.91 | 95.22 | 94.89 | 71.76 | 80.41 | 89.02 | 33.77 | 75.30 | 63.91 | 71.79 | 70.71 | 75.02 |
| VAE + disentangled (SpecPL) HM | **73.34** | 95.31 | **95.79** | **73.06** | **82.14** | **89.28** | **35.98** | 75.17 | **64.06** | **73.83** | **75.03** | **76.52** |

*Table 8.* **Per-dataset teacher/decomposition ablation.** We report HM for each dataset. The best result in each column is highlighted in bold.

Although the raw counterfactual loss remains numerically larger, it also decreases from 4.8723 to 3.7582. With $\lambda_3 = 0.1$, its effective contribution is reduced from 0.4872 to 0.3758. These trends indicate that counterfactual granule supervision behaves as a regularizer rather than dominating the main classification objective.

**Teacher choice and explicit disentanglement.** We further separate two factors in SpecPL: the teacher representation space and the explicit Base/Detail decomposition. Table 7 reports the average Base, Novel, and HM results over the 11-dataset benchmark, while Table 8 provides the corresponding per-dataset HM values.

The results show that the performance gain is not explained by the VAE teacher alone. Using CLIP visual features as the teacher while keeping the explicit low/high decomposition already improves the average HM from 71.66 to 75.02, demonstrating that the disentanglement mechanism itself is beneficial. Conversely, replacing CoOp with a VAE-based holistic teacher without explicit Base/Detail separation also improves HM to 75.23, indicating that the teacher representation space matters. Combining both factors yields the best trade-off, reaching 76.52 average HM and 70.74 average Novel accuracy. Notably, this configuration does not obtain the highest Base accuracy, suggesting that the improvement mainly

| | Base | | | Novel | | | HM | | |
|---|---|---|---|---|---|---|---|---|---|
| Shot | CoOp | SpecPL | Δ | CoOp | SpecPL | Δ | CoOp | SpecPL | Δ |
| 1-shot | 72.51 | 73.65 | +1.14 | 66.99 | 70.56 | **+3.57** | 69.64 | 72.07 | **+2.43** |
| 2-shot | 76.22 | 76.64 | +0.42 | 66.65 | 69.79 | **+3.14** | 71.12 | 73.06 | **+1.94** |
| 4-shot | 77.92 | 78.94 | +1.02 | 65.28 | 70.11 | **+4.83** | 71.04 | 74.26 | **+3.22** |
| 8-shot | 80.59 | 81.08 | +0.49 | 65.84 | 69.82 | **+3.98** | 72.47 | 75.03 | **+2.56** |
| 16-shot | 82.69 | 83.32 | +0.63 | 63.22 | 70.74 | **+7.52** | 71.66 | 76.52 | **+4.86** |

*Table 9.* **Few-shot base-to-novel average results.** Average Base, Novel, and HM results across 1/2/4/8/16-shot settings on the 11-dataset base-to-novel benchmark.

| Method | Shot | ImageNet | Caltech101 | OxfordPets | StanfordCars | Flowers102 | Food101 | FGVCAircraft | SUN397 | DTD | EuroSAT | UCF101 | Average |
|---|---|---|---|---|---|---|---|---|---|---|---|---|---|
| CoOp | 8-shot | 72.84 | 95.05 | 94.21 | 70.15 | 75.17 | 88.82 | 25.35 | 70.78 | 59.43 | 69.55 | 69.35 | 72.47 |
| | 16-shot | 71.92 | 93.73 | 94.47 | 68.13 | 74.06 | 85.19 | 28.75 | 72.51 | 54.24 | 68.69 | 67.46 | 71.66 |
| | Δ | **-0.92** | **-1.32** | +0.26 | **-2.02** | **-1.11** | **-3.63** | +3.40 | +1.73 | **-5.19** | -0.86 | **-1.89** | **-0.81** |
| SpecPL | 8-shot | 72.84 | 95.55 | 95.30 | 71.86 | 80.41 | 89.13 | 32.61 | 74.55 | 60.63 | 78.30 | 71.81 | 75.03 |
| | 16-shot | 73.34 | 95.31 | 95.79 | 73.06 | 82.14 | 89.28 | 35.98 | 75.17 | 64.06 | 73.83 | 75.03 | 76.52 |
| | Δ | **+0.50** | -0.24 | **+0.49** | **+1.20** | **+1.73** | **+0.15** | **+3.37** | **+0.62** | **+3.43** | -4.47 | **+3.22** | **+1.49** |

*Table 10.* **HM stability from 8-shot to 16-shot.** CoOp often loses HM when increasing from 8 to 16 shots, whereas SpecPL is generally more stable and improves on average.

comes from better base-to-novel transfer rather than stronger fitting to base classes.

**Few-shot behavior.** We evaluate few-shot behavior under two complementary protocols. The main base-to-novel protocol trains prompts with $K$ samples per base class and reports Base, Novel, and harmonic mean (HM) accuracy, thereby exposing the trade-off between base-class fitting and transfer to held-out novel classes. In contrast, the conventional all-class protocol samples $K$ examples from every class and trains/evaluates over the full label space; since all categories are seen during training, it primarily reflects in-distribution sample efficiency rather than base-to-novel generalization.

Table 9 summarizes the average base-to-novel results, while Table 10 reports per-dataset HM changes from 8-shot to 16-shot. Figures 8–10 provide visual diagnostics: Figure 8 shows HM curves under the base-to-novel split, Figure 9 decomposes them into Base, Novel, and HM trends, and Figure 10 reports the all-class few-shot results.

SpecPL consistently improves HM across all evaluated base-to-novel shot settings, with gains of +2.43 to +4.86 points from 1-shot to 16-shot. The improvement is particularly pronounced on novel-class accuracy, where the gains reach +3.57 to +7.52 points, indicating that SpecPL enhances transfer to unseen categories rather than merely improving base-class fitting. In the higher-shot regime, CoOp's average HM drops from 72.47 at 8-shot to 71.66 at 16-shot, whereas SpecPL further improves from 75.03 to 76.52. The per-dataset results in Table 10 show similar trends on most datasets, suggesting that spectral granularity regularizes prompt learning when more base-class samples are available. Figure 10 reports the complementary all-class few-shot protocol, where all categories are visible during training. This protocol is useful for comparing sample efficiency, but should be interpreted separately from the base-to-novel setting, as it does not assess transfer to held-out novel classes. Thus, Figure 10 serves as the standard few-shot reference, while Figures 8–9 and Tables 9–10 provide a more diagnostic evaluation of base-class fitting, novel-class transfer, and their balance.

**Efficiency.** Finally, we quantify the additional cost introduced by SpecPL. Table 11 compares MMRL and MMRL + SpecPL, and Table 12 reports the one-time VAE cache construction cost across datasets.

SpecPL introduces a modest number of additional trainable parameters and a one-time VAE preprocessing step, but it does not add an inference-time VAE or FiLM branch. On MMRL, learnable parameters increase from 4.992M to 8.147M, while optimization time increases only from 335.47s to 340.35s. The main additional cost is the one-time VAE cache construction, which takes 111.73s and stores 35.77MB on ImageNet. At inference, FLOPs remain unchanged at 34.2121 GFLOPs, and the total test time on 25k images increases marginally from 53.39s to 53.90s. Therefore, SpecPL provides a complementary accuracy gain with negligible inference-side overhead.

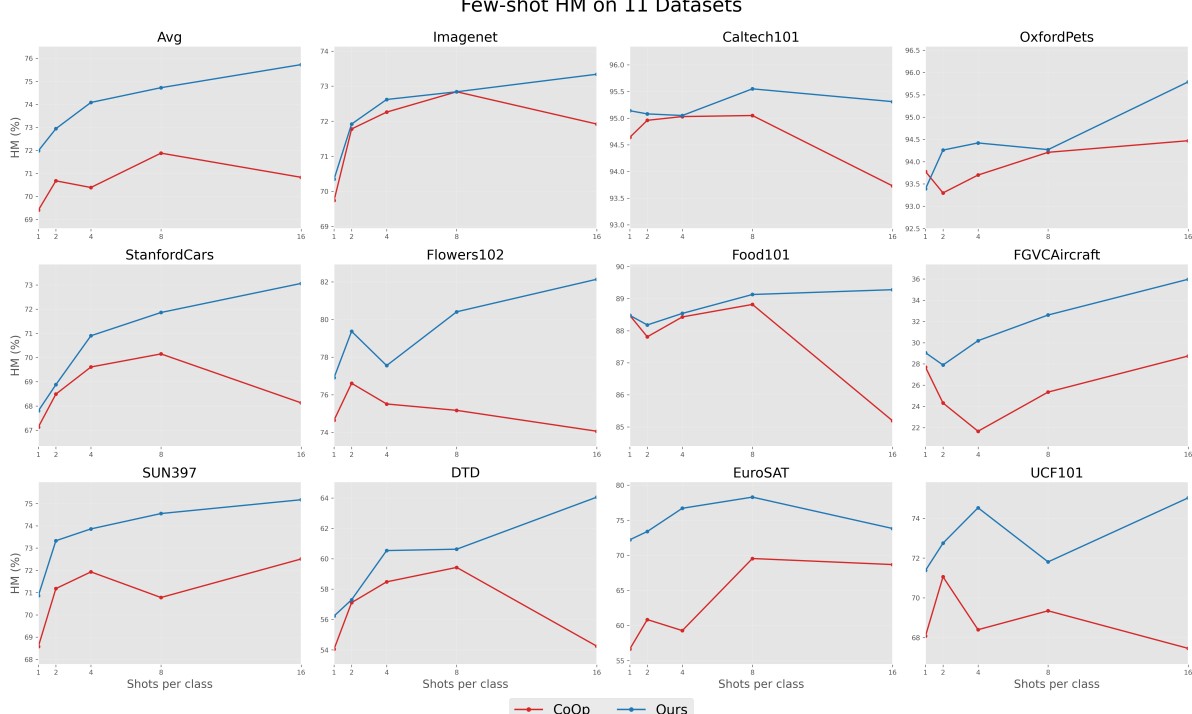

*Figure 8.* **Few-shot HM under the base-to-novel protocol on 11 datasets.** HM curves for CoOp and CoOp + SpecPL under 1/2/4/8/16-shot settings.

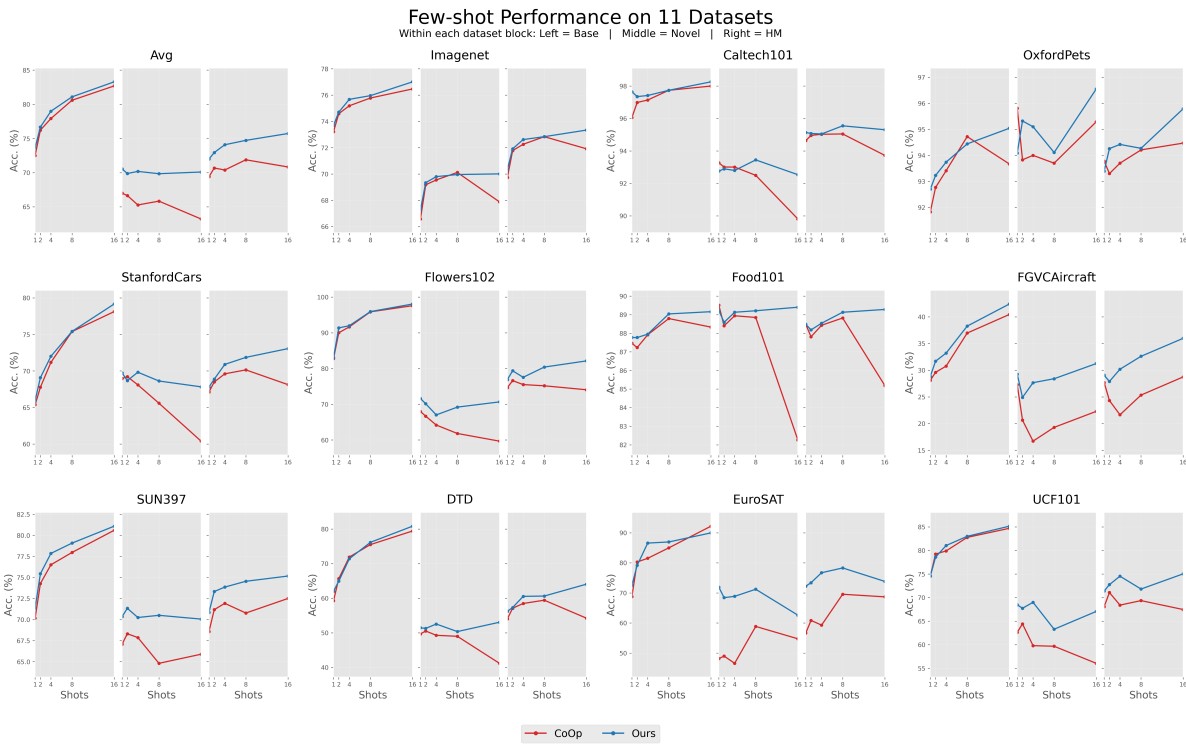

*Figure 9.* **Few-shot Base/Novel/HM performance under the base-to-novel protocol on 11 datasets.** We report full Base, Novel, and HM curves to show how SpecPL affects both base-class fitting and novel-class transfer.

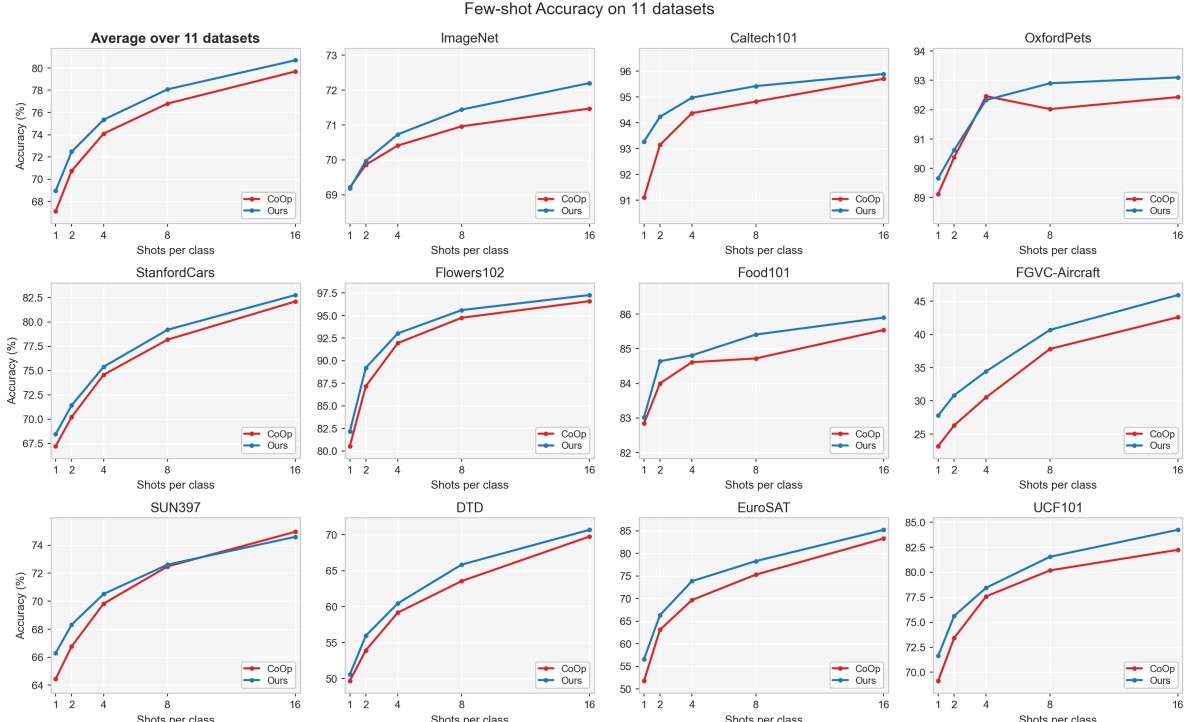

*Figure 10.* **Conventional all-class few-shot performance on 11 datasets.** Unlike the base-to-novel setting in Figures 8 and 9, this protocol samples few-shot training data from all categories and evaluates on the full label space. We report all-class performance curves for CoOp and CoOp + SpecPL under 1/2/4/8/16-shot settings.

| Method | HM Gain over MMRL | Train | | | | | Infer | |
| | | Learnable Params. | One-time Preproc. | Optim. Time (s) | GPU Hours | Extra Cache | Test Time (25k imgs, s) | FLOPs (GFLOPs) |
|---|---|---|---|---|---|---|---|---|
| MMRL | – | 4.992 M | 0 | 335.47 | 0.0932 | – | 53.39 | 34.2121 |
| MMRL + SpecPL | +0.31 | 8.147 M | 111.73 s | 340.35 | 0.0945 | 35.77 MB | 53.90 | 34.2121 |
| Relative Change | – | +3.155 M | one-time only | +1.45% | +1.39% | 35.77 MB only | +0.96% | $\approx 0.00\%$ |

*Table 11.* **Efficiency comparison.** We compare MMRL and MMRL + SpecPL in terms of train-time overhead, inference cost, learnable parameters, and cache size.

| Metric | ImageNet | Caltech101 | OxfordPets | StanfordCars | Flowers102 | Food101 | FGVCAircraft | SUN397 | DTD | EuroSAT | UCF101 |
|---|---|---|---|---|---|---|---|---|---|---|---|
| Preproc. Time (s) | 111.73 | 11.02 | 6.47 | 21.65 | 13.90 | 11.29 | 11.33 | 43.97 | 5.66 | 1.55 | 11.30 |
| Teacher Samples | 8,000 | 800 | 348 | 1,568 | 816 | 816 | 800 | 3,184 | 384 | 80 | 816 |
| Cache Size (MB) | 35.77 | 3.58 | 1.36 | 7.03 | 3.65 | 3.65 | 3.58 | 14.22 | 1.73 | 0.37 | 3.65 |

*Table 12.* **One-time VAE cache preprocessing cost.** We report preprocessing time, number of teacher samples, and cache size for each dataset.

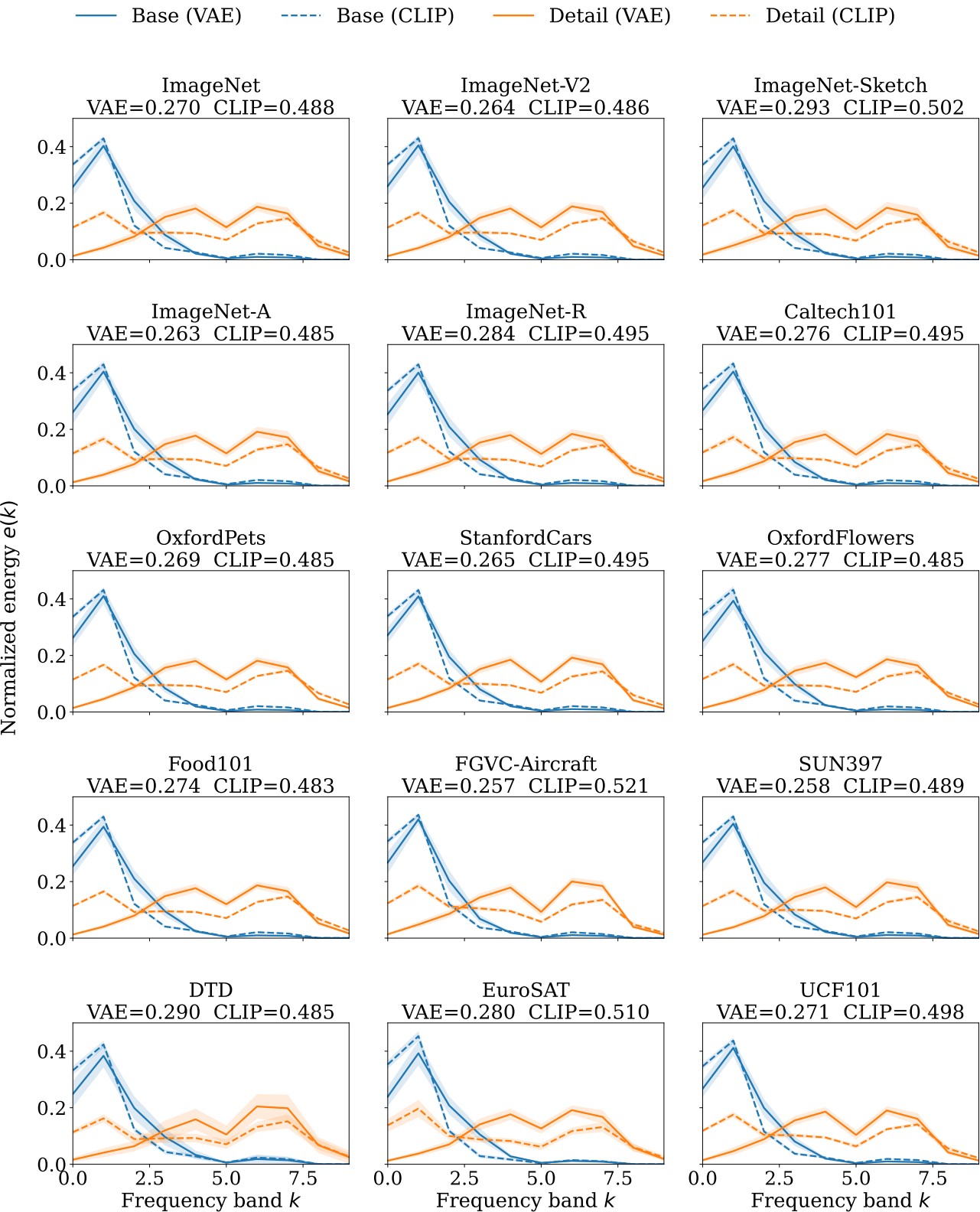

*Figure 11.* **Spectral energy curves across all datasets.** Each subplot reports the overlap scores (VAE/CLIP) in the title; one shared legend is shown on top.

