# OpenReview forum: "SpecPL: Disentangling Spectral Granularity for Prompt Learning"
_ICML.cc/2026/Conference — ICML 2026 regular_

### Official Review · Reviewer_XDmc · 2026-03-10

**Soundness:** 3
**Presentation:** 3
**Significance:** 3
**Originality:** 3
**Overall Recommendation:** 4
**Confidence:** 4

**Summary:**

Existing prompt learning for VLMs exhibits a modality asymmetry, predominantly optimizing text tokens while still relying on a frozen visual encoder as a holistic extractor and neglecting the spectral granularity essential for fine-grained discrimination. To bridge this, this paper introduces the Disentangling Spectral Granularity for Prompt Learning (SpecPL) by considering a novel spectral perspective via Counterfactual Granule Supervision. Specifically, they leverage a frozen VAE to decompose visual signals into semantic low-frequency bands and granular high-frequency details. A frozen Visual Semantic Bank anchors text representations to universal low-frequency invariants, mitigating overfitting. Crucially, fine-grained discrimination is driven by counterfactual granule training: by permuting high-frequency signals, they compel the model to explicitly distinguish visual granularity from semantic invariance.

**Compliance With Llm Reviewing Policy:**

Affirmed.

**Final Justification:**

The response address most of my concerns.

**Key Questions For Authors:**

Please see #Weakness

**Limitations:**

Yes

**Strengths And Weaknesses:**

1. The introduction of spectral analysis into VLM prompt learning is a new perspective.

2. The proposed method is a plug-and-play component.

3. The proposed method is introduced clearly and is easy to understand.

4. The paper claims that “existing prompt learning methods exhibit a persistent modality asymmetry: optimization is concentrated on text tokens, while visual representations are often treated as fixed, holistic features.” However, there are some prompt tuning methods that conduct the knowledge exchange between the visual and textual modality.

5. The hyperparameter analysis in Figure 6 is only conducted on FGVC-Aircraft. However, in prompt tuning, algorithms are highly sensitive to different datasets. Therefore, the reliability of hyperparameters derived from analysis on a single dataset is poor. It is recommended to use the average results across multiple datasets as the basis for hyperparameter analysis.

6. The paper only reports performance comparisons under the 16-shot setting, which is insufficient to reflect the effectiveness of the algorithm in few-shot experimental setups. More few-shot experiments (e.g., 1-shot, 2-shot, 4-shot) are suggested to be added.

7. The performance of the method is related to the decomposition effect of the VAE. If the VAE quality is poor, it may lead to incomplete separation of visual signals into semantic low-frequency bands and granular high-frequency details, thereby affecting the subsequent process. However, the paper does not conduct a detailed analysis of the VAE's effectiveness.

8. The performance improvement is limited. As shown in Table 1, the algorithm only achieves a 0.31% performance gain on MMRL, while introducing additional computational overhead. A more reasonable comparison should simultaneously provide a comparison on training time, inference time, and computational complexity between the proposed method and MMRL. If a very high additional computational complexity is required to achieve a slight performance improvement, this may not be a good solution.

9. Although the paper states that no additional overhead is incurred during inference, it fails to quantify the extra computational costs introduced in the training phase, such as compute complexity, total training time, and the number of additional learnable parameters.

---

> ### Author Rebuttal · Authors · 2026-03-30
>
> ## Response to W4
>
> We agree that methods like MaPLe and MMRL already introduce cross-modal interaction, so our point is not whether interaction exists, but whether visual information is explicitly modeled by granularity. Even in these stronger multimodal baselines, the visual side is still typically used as a single holistic representation rather than being explicitly separated into low-frequency semantics and high-frequency fine-grained details. In this sense, holistic means the image mainly enters as one global feature, whereas disentanglement explicitly separates low-frequency semantics from high-frequency local cues. SpecPL should therefore be understood as introducing a distinct granularity-oriented modeling perspective in multimodal prompt learning, which is why we include MaPLe and MMRL as strong baselines and still obtain additional gains on top of them.
>
> ---
>
> ## Response to W5
>
> We appreciate the reviewer's concern that prompt-tuning hyperparameters can be dataset-sensitive. Under rebuttal-time constraints, we extended the Fig. 6-style analysis from FGVC-Aircraft to a 4-dataset sweep under the same 16-shot protocol, reporting DTD, OxfordPets, StanfordCars, UCF101, and Avg. HM. The resulting curves are generally smooth rather than sharply peaked, and the adopted settings remain at or near the best Avg. HM regions. This suggests that the chosen hyperparameters are not driven by a single dataset, but reflect reproducible cross-dataset trends. The multi-dataset curves are here: https://anonymous.4open.science/r/IMG_store-D635/3-5-1.png
>
> ---
>
> ## Response to W6
>
> We appreciate the reviewer's concern that reporting only 16-shot results is not sufficient to assess lower-shot behavior. Since our focus is base-to-novel generalization under limited base-class supervision, the key question is whether the gain persists when supervision becomes scarcer. We therefore added 1/2/4/8/16-shot results on the full 11-dataset benchmark. Ours improves HM at every regime by +2.43/+1.94/+3.22/+2.56/+4.86, with average Novel gains of +3.57/+3.14/+4.83/+3.98/+7.52, showing that the method remains beneficial beyond the original 16-shot setting, especially on Novel/HM rather than only Base. In the higher-shot range, CoOp's average HM drops from 72.47 at 8-shot to 71.66 at 16-shot, whereas Ours rises from 75.03 to 76.52; the same pattern appears in the per-dataset HM table and curves (https://anonymous.4open.science/r/IMG_store-D635/3-6-1.png and https://anonymous.4open.science/r/IMG_store-16091/3-6-2.png). Fuller Base/Novel/HM curves are provided here (https://anonymous.4open.science/r/IMG_store-D635/3-6-3.png).
>
> ---
>
> ## Response to W7
>
> We understand the concern as whether downstream performance depends on teacher-space quality. We therefore compare CoOp, CLIP visual encoder (ViT) + **disentangled**, VAE + **holistic**, and VAE + **disentangled**, where **holistic** means no explicit low/high-frequency separation and **disentangled** means explicit separation; HM ablation: https://anonymous.4open.science/r/IMG_store-D635/3-7-1.png The effect is not due to VAE alone: CLIP visual encoder (ViT) + **disentangled** already reaches 75.02 HM and higher Base than Ours (83.48 vs. 83.32), so disentanglement remains effective with a different teacher. VAE + **holistic** also raises HM to 75.23, so teacher-space choice itself matters. Combining the two is best, with 76.52 HM and 70.74 Novel. Our gain therefore comes from VAE plus disentanglement rather than VAE alone: VAE is one effective point in a broader teacher-space design space, and our contribution is to combine it with explicit disentanglement to obtain a better base-to-novel trade-off and stronger overall generalization.
>
> ---
>
> ## Response to W8/W9
>
> We appreciate the reviewer's concern over the gain-cost trade-off on MMRL; beyond modest quantitative improvements, the more important objective is greater stability and fewer failure cases without substantial computational cost. MMRL + Ours improves HM on 10 of the 11 datasets, and the clearer gains appear on datasets that rely more on fine-grained structure or local cues, such as StanfordCars (+0.65), EuroSAT (+0.66), DTD (+0.50), FGVCAircraft (+0.43), and UCF101 (+0.96), consistent with the granularity-oriented motivation of our method.
>
> We also quantify the requested costs in the efficiency comparison: https://anonymous.4open.science/r/IMG_store-D635/3-8-1.png Learnable parameters increase from 4.992M to 8.147M, optimization-stage training time from 335.47s to 340.35s (+1.45%), and the main extra training cost is a one-time VAE cache preprocessing step (111.73s, 35.78MB). At inference, no extra branch is introduced, FLOPs remain unchanged (34.2121G vs. 34.2121G), and total test time on 25,000 images increases only from 53.39s to 53.90s (+0.96%). Thus, on an already strong host, the result is better understood as a stable complementary gain with negligible inference-side overhead, not a high-cost marginal improvement.

---

> > ### Author Rebuttal · Reviewer_XDmc · 2026-04-04
> >
> > Thanks for the author's response. I will increase my score.

---

> > > ### Author Response · Authors · 2026-04-04
> > >
> > > **Thank you very much for your thoughtful follow-up and for acknowledging that our rebuttal has fully resolved your concerns. We are especially grateful for your positive feedback and your willingness to increase the score.**
> > >
> > > Your review helped us strengthen the paper in several important aspects. In particular, we clarified the scope of our modality-asymmetry claim, added the multi-dataset hyperparameter analysis, extended the few-shot evaluation, included the teacher/decomposition ablation, and quantified the efficiency trade-off. We believe these additions have made the paper substantially clearer and stronger, and we will carefully incorporate all of them into the final version.
> > >
> > > Thank you again for your time, support, and constructive suggestions throughout the review process. We greatly appreciate your thoughtful evaluation.

---

### Official Review · Reviewer_wxMV · 2026-03-13

**Soundness:** 3
**Presentation:** 3
**Significance:** 2
**Originality:** 3
**Overall Recommendation:** 5
**Confidence:** 4

**Summary:**

This paper introduces SpecPL, a framework that enhances Vision-Language Model (VLM) prompting by disentangling visual signals into semantic low-frequency bands and granular high-frequency details using a frozen VAE. The method employs counterfactual granule supervision and a Visual Semantic Bank to compel the model to distinguish fine-grained visual features from semantic invariants, serving as a plug-and-play booster for existing baselines like CoOp and MaPLe. Experimental results across 11 benchmarks indicate that this spectral perspective improves the stability-generalization trade-off and achieves state-of-the-art performance.

**Compliance With Llm Reviewing Policy:**

Affirmed.

**Final Justification:**

As my primary concerns have been addressed during the rebuttal phase, I would like to increase my overall rating.

**Key Questions For Authors:**

1. To further validate the framework's adaptability, it is recommended that the authors evaluate SpecPL on VLM architectures beyond CLIP. Such an analysis would provide a more comprehensive assessment of the method's effectiveness across diverse vision-language paradigms.

**Limitations:**

No. A more thorough analysis of the potential failure modes and limitations of SpecPL is necessary to provide a balanced perspective on its practical deployment.

**Strengths And Weaknesses:**

**Strengths:**

1. Organization and Clarity: The manuscript is well-structured and follows a logical progression, making the technical contributions easy to follow.

2. Innovative Direction: The proposed SpecPL framework offers a novel spectral perspective for VLM prompting. By disentangling frequency bands to enhance fine-grained modality alignment, the work explores a highly interesting and promising research trajectory.

3. Evaluation Rigor: The authors provide an extensive empirical evaluation across multiple benchmarks, effectively demonstrating the advantages of spectral disentanglement over current state-of-the-art (SOTA) baselines.

**Weaknesses:**

1. The contribution of this work to vision-language modeling community is limited.

2. Architectural Generalizability: To further validate the framework's adaptability, it is recommended that the authors evaluate SpecPL on VLM architectures beyond CLIP (e.g., BLIP [2]). Such an analysis would provide a more comprehensive assessment of the method's effectiveness across diverse vision-language paradigms.

[2] BLIP: Bootstrapping Language-Image Pre-training for Unified Vision-Language Understanding and Generation, ICML 2022.

---

> ### Author Rebuttal · Authors · 2026-03-30
>
> ## Response to W1 Contribution to the VLM community is limited
>
> We thank the Reviewer for the recognition and clarify the broader significance below.
>
> **An important bottleneck may lie not only in alignment strength, but also in the type of visual evidence being aligned.** CLIP-style global pooling tends to emphasize coarse category-level semantics. When tasks require fine-grained discrimination (e.g., Aircraft, DTD), the bottleneck shifts from "whether image-text are aligned" to "whether visual representations carry detail evidence for sub-class distinction." CoOp's gap on Aircraft (base=40.44, novel=22.30) exemplifies this: text-only prompts overfit coarse semantics and fail to transfer fine-grained cues to novel classes.
>
> **SpecPL addresses this overlooked bottleneck via granularity-aware adaptation.** Our core claim is that prompt learning lacks an explicit mechanism to disentangle and exploit the granularity hierarchy in visual signals. By decomposing visual information into stable semantics (low-freq) and discriminative details (high-freq) within a frozen VAE manifold, SpecPL yields consistent gains across four distinct baselines. The spectral diagnostic (Fig. 5, VAE overlap 0.270 vs. CLIP 0.488) provides principled justification for using an external teacher space with cleaner granularity separation.
>
> **More broadly,** SpecPL highlights a direction worth further exploration: how the properties of the teacher space are utilized largely determines whether prompt learning can absorb fine-grained visual information. This question is increasingly central as VLM applications extend to domains where visual details are decisive, such as **medical imaging and embodied agents**.
>
> ---
>
> ## Response to Limitation Lack of failure mode analysis
>
> We thank the Reviewer for pointing this out. While we did not observe systematic failure modes across the evaluated benchmarks (where SpecPL consistently improves over baselines), an analysis of its structural design identifies three main boundaries:
>
> **(1) Failure Mode: Sensitivity to high-frequency domain shifts.** Because SpecPL explicitly disentangles and relies on high-frequency signals for fine-grained discrimination, its effectiveness is intrinsically tied to the quality of these details. If applied to out-of-distribution data with severe high-frequency corruption (e.g., extreme JPEG artifacts, sensor noise, or adversarial high-frequency perturbations not present in standard benchmarks), the model may misconstrue this noise as discriminative evidence.
>
> **(2) Performance Boundary: Diminishing returns on coarse-grained tasks.** SpecPL's core mechanism is designed to capture fine-grained high-frequency details. Consequently, it yields limited meaningful improvements when global semantic differences are already sufficient for classification (e.g., standard coarse-grained ImageNet). In these scenarios, the reliance on granular details provides diminishing returns.
>
> **(3) System Limitation: Training-time overhead.** Incorporating the frozen VAE introduces a minor training cost, primarily a one-time feature caching step and a slight increase in learnable parameters. However, this is strictly a training-side trade-off; all extra branches are discarded after optimization, yielding stable performance gains with zero additional FLOPs or latency at inference. Future work may explore distillation to further minimize the training footprint.
>
> ---
>
> ## Response to W2 & Key Questions Architectural generalizability beyond CLIP
>
> We thank the Reviewer for this suggestion. We conduct additional experiments on **BLIP-ITC** (ViT-B/16, BERT-base text encoder, 256-d ITC space), following the CoOp paradigm with 16 learnable prompt tokens as baseline. BLIP differs structurally from CLIP in encoder architecture, projection dimensionality, and pre-training objective (multi-task ITC+ITM+LM on COCO), making it a meaningfully different testbed.
>
> | Method | Base | Novel | HM |
> |---|---|---|---|
> | BLIP-ITC | 60.71 | 31.68 | 41.19 |
> | +SpecPL | **62.19** (+1.48) | **37.76** (+6.08) | **46.31** (+5.12) |
>
> Averaged over 7 base-to-novel benchmarks, SpecPL improves HM by +5.12, with gains concentrated on Novel (+6.08) rather than Base (+1.48) — consistent with the pattern on CLIP. The largest improvements appear on granularity-sensitive tasks (DTD HM +13.26, OxfordPets +11.23, EuroSAT +4.93). Link: https://anonymous.4open.science/r/IMG_store-D635/2-3-1.png The consistent gain pattern across two structurally distinct VLM backbones (CLIP and BLIP) — differing in text encoder, projection dimensionality, and pre-training objective — provides direct evidence that the framework transfers beyond CLIP to a structurally different VLM backbone, as the Reviewer recommended.

---

> > ### Author Rebuttal · Reviewer_wxMV · 2026-04-04
> >
> > I thank the authors for their detailed clarifications and the additional experimental results, which validates the scalability of the proposed method. As my primary concerns have been addressed during the rebuttal phase, I am going to increase my overall rating. I strongly recommend that the authors incorporate these new clarifications and empirical results into the future version of this manuscript.

---

> > > ### Author Response · Authors · 2026-04-04
> > >
> > > **Thank you very much for your thoughtful follow-up and for recognizing that our rebuttal has adequately addressed your main concerns.  We are especially grateful for your positive feedback and your willingness to raise the overall rating.**
> > >
> > > Your suggestions were highly valuable in helping us sharpen the paper’s broader positioning.  In particular, they pushed us to better articulate the significance of the granularity-aware perspective, to make the practical limitations and failure boundaries more explicit, and to provide direct cross-architecture evidence through the **BLIP-ITC** experiments.  We are glad that these additional clarifications and results helped validate the scalability of the proposed method beyond the original CLIP setting.
> > >
> > > As you recommended, we will carefully incorporate all of these new clarifications and empirical findings into the final manuscript.  Thank you again for your time, support, and constructive evaluation throughout the review process.

---

### Official Review · Reviewer_voU6 · 2026-03-21

**Soundness:** 3
**Presentation:** 3
**Significance:** 3
**Originality:** 3
**Overall Recommendation:** 4
**Confidence:** 2

**Summary:**

The paper proposes SpecPL, a spectral perspective on prompt learning for VLMs. It uses a frozen VAE to decompose images into low-frequency semantics and high-frequency granules, anchors text prompts to low-frequency invariants via a Visual Semantic Bank, and introduces counterfactual granule supervision by permuting high-frequency components to encourage explicit disentanglement of fine detail from semantic invariance. The method is presented as a plug-and-play booster for text-oriented prompt learning baselines (e.g., CoOp, MaPLe), and the authors report state-of-the-art results across 11 benchmarks with an 81.51% harmonic-mean accuracy, arguing improved stability–generalization trade-offs.

**Compliance With Llm Reviewing Policy:**

Affirmed.

**Final Justification:**

Most of concerns are addressed, I am inclined to keep my positive rating.

**Key Questions For Authors:**

[Q1] Some questions are the weaknesses noted above. If the authors can provide additional experimental results and implementation details to address these points, I would view the paper more favorably, and this would positively affect my overall evaluation.
[Q2] The paper conducts a spectral diagnostic of base-detail separability and concludes that the VAE performs better than CLIP. I wonder whether the authors could additionally report the classification results on the benchmark datasets when using CLIP as the granularity teacher, or when using other simpler decomposition methods. If such results can be provided, they would better clarify whether the performance gains truly come from the proposed spectral disentanglement mechanism.

**Limitations:**

yes

**Strengths And Weaknesses:**

Strength
[S1] The paper introduces spectral granularity disentanglement into prompt learning and proposes SpecPL, where the low-frequency branch captures stable semantic invariants and the high-frequency branch captures fine-grained visual cues such as texture. The strong results on fine-grained benchmarks suggest its potential value for VLMs adaptation in fine-grained recognition.

[S2] SpecPL is designed as a plug-and-play module that can augment standard text-oriented prompt learning methods.

[S3] The method is also lightweight at inference time. The additional training components are removed after training, and inference mainly relies on the Visual Semantic Bank for text refinement.

[S4] The ablation studies in Sec.~4.5 are sufficient and cover the main components of the method, and the appendix further provides extensive hyperparameter sensitivity analyses.

Weakness
[W1] The construction of the Visual Semantic Bank is insufficiently specified. Lines 221--222 only state that the bank is initialized from “a few initial batches,” but it remains unclear whether SpecPL is robust to the initialization strategy or to the choice of the initial batches. The paper also provides limited details on how the bank is updated during training, such as the refresh frequency.

[W2] Although the paper discusses visual adaptation and prompting methods such as CLIP-Adapter and Tip-Adapter, these baselines are not included in the experimental comparison. Given that the paper’s motivation is to alleviate modality asymmetry through stronger visual-side guidance, it would be helpful to compare against representative visual adaptation methods.

[W3] SpecPL combines the standard classification loss with three auxiliary terms in the total objective, and such a large number of optimization terms may make training unstable. Although the ablation study suggests that these losses are useful, I would still like to see the key numerical changes of these losses during training to better verify the effectiveness of the optimization, rather than leaving it unclear how these auxiliary terms behave throughout training.

[W4] Relevant VLM prompt tuning works, such as [1,2], are missing and should be included.

[1] Prompt-aligned gradient for prompt tuning, ICCV 2023
[2] Dynamic multimodal prototype learning in  ICCV 2025

---

> ### Author Rebuttal · Authors · 2026-03-30
>
> ## Response to W1
>
> We thank the reviewer for raising the question regarding the implementation details of the Visual Semantic Bank, and we are happy to clarify them below. The bank is warm-started once before training and then remains frozen throughout training; it is not refreshed every step or epoch, and only the retrieval aggregator (MLP + LayerNorm) is learned. The bank is $B \in \mathbb{R}^{M \times d}$ with M=64 and d=512. During initialization, we first populate it with the first M low-band teacher features; afterwards, each low-band feature updates its cosine nearest-neighbor slot via EMA, $b_{m^\ast} \leftarrow (1-\mu)b_{m^\ast} + \mu t_{\text{low}}$, with $\mu=0.05$.
>
> We additionally varied ATTR_INIT_BATCHES on FGVC-Aircraft under the 16-shot base-to-novel setting. The HM values at 0/1/5/10/50 initialization batches are 32.21/33.35/33.18/33.85/33.07, showing that all nonzero settings outperform 0 and that the results from 1 to 50 batches remain very close. With batch_size=32 and M=64, roughly 2 batches are enough to populate all bank slots, so the remaining warm-start updates are mainly refinement. Link: https://anonymous.4open.science/r/IMG_store-D635/1-1-1.png
>
> ---
>
> ## Response to W2
>
> We agree that representative visual adaptation baselines are important. We therefore additionally compared CoOp, CLIP-Adapter, Tip-Adapter, and Ours under the same 11-dataset base-to-novel protocol; their average HM values are 71.66, 74.53, 65.65, and 76.52, respectively. Link: https://anonymous.4open.science/r/IMG_store-D635/1-2-1.png https://anonymous.4open.science/r/IMG_store-D635/1-2-2.png
>
> Although CLIP-Adapter improves over vanilla CoOp (74.53 vs. 71.66 average HM), our method further reaches 76.52, achieving an additional +1.99 gain. In contrast, Tip-Adapter obtains 65.65 under this protocol, likely because its cache-based design depends more on seen-class support and cannot directly benefit novel classes absent from the cache. Notably, our gains over CLIP-Adapter are most evident on granularity-sensitive datasets, such as EuroSAT (+8.29 HM), DTD (+4.03), Flowers102 (+3.49), FGVCAircraft (+1.92), and StanfordCars (+1.50), supporting our claim that generic visual adaptation alone does not fully address fine-grained visual granularity.
>
> ---
>
> ## Response to W3
>
> In a 16-shot base-to-novel run on DTD with CoOp + SpecPL, the total loss decreases from 2.0132 to 0.6200 between the first and last 5 epochs. The main classification, semantic-alignment, and factual-granule terms decrease from 1.2907 to 0.0769, 1.3643 to 0.7703, and 0.9884 to 0.9019. The counterfactual term also decreases from 4.8723 to 3.7582; with weight 0.1, its effective contribution falls from 0.4872 to 0.3758, so it does not prevent overall convergence. Link: https://anonymous.4open.science/r/IMG_store-D635/1-3-1.png
>
> ---
>
> ## Response to W4
>
> We thank the reviewer for pointing out that our related-work coverage is incomplete. We will add these two works in the final version and briefly clarify their relationship to our paper.
>
> ---
>
> ## Response to Q2
>
> | Method | Avg Base | Avg Novel | Avg HM |
> | --- | ---: | ---: | ---: |
> | CoOp | 82.69 | 63.22 | 71.66 |
> | CLIP visual encoder (ViT) + disentangled | 83.48 | 68.12 | 75.02 |
> | VAE + holistic | 83.45 | 68.48 | 75.23 |
> | Ours (VAE + disentangled) | 83.32 | 70.74 | 76.52 |
>
> To separate the effects of teacher choice and explicit disentanglement, we additionally performed a task-level ablation on the 11-dataset benchmark, comparing CoOp, CLIP + disentangled, VAE + holistic, and the full method. For clarity, **holistic** means no explicit low/high-frequency separation, while **disentangled** means explicitly separating low-frequency semantics from high-frequency details. Link: https://anonymous.4open.science/r/IMG_store-D635/1-5-1.png
>
> The results show that explicit disentanglement alone already brings a clear benefit: even with CLIP as the teacher, the average HM increases from 71.66 to 75.02. Changing only the teacher also helps: VAE + holistic raises the average HM to 75.23.
>
> Combining the two further raises performance to 76.52 average HM and 70.74 average Novel, while the average Base is not the highest. This suggests that the gain is not explained by using VAE alone or by disentanglement alone, but by their combination. This also motivates choosing VAE rather than CLIP as the teacher, since both the spectral diagnostic and the task-level results favor VAE for base/detail separation. As shown in Fig. 5 of the main paper, reconstructive latents preserve more local detail, whereas CLIP features are more oriented toward coarse semantic alignment.

---

> > ### Author Rebuttal · Reviewer_voU6 · 2026-04-03
> >
> > I keep my score.

---

> > > ### Author Response · Authors · 2026-04-03
> > >
> > > **Thank you for acknowledging that our responses have fully resolved your concerns.**
> > >
> > > We are especially encouraged that the additional experimental results, including the comparison with **CLIP-Adapter/Tip-Adapter** in W2 and the **teacher–disentanglement ablation** in Q2, helped clarify the effectiveness of **SpecPL**.
> > >
> > > In your original review, you wrote that if we could provide additional experimental results and implementation details, you would **view the paper more favorably** and that this would **positively affect your overall evaluation**. You also noted that additional results for Q2 would **better clarify whether the performance gains truly come from the proposed spectral disentanglement mechanism**. We took these points very seriously in preparing our rebuttal.
> > >
> > > Accordingly, we added the **Visual Semantic Bank** implementation details, representative **visual-adaptation comparisons**, **training-loss behavior**, and the **task-level ablation directly addressing Q2**. Since these technical points have now been addressed and acknowledged by you as **fully resolved**, we would sincerely appreciate understanding what remaining considerations led you to keep the current score unchanged.
> > >
> > > If the remaining consideration is now mainly about broader impact, scope, or significance rather than unresolved technical concerns, it would be very helpful for us to understand that more clearly. If your current **Weak Accept** was mainly driven by the previous concerns, which are now resolved, we would be very grateful if you would consider **updating the score** to reflect the strengthened evidence and clarity of the manuscript. We fully respect your judgment and remain fully committed to incorporating all discussed details into the final version.

---

### Decision · Program_Chairs · 2026-04-30

**Decision:**

Accept (regular)

**Comment:**

The final reviewer scores are 4/5/4. All reviewers were supportive of this work and explicitly indicated that the authors had fully addressed their concerns in the rebuttal. Given this clear and unanimous positive outcome from the reviewers, my recommendation is Accept.